# PoMtVRS: Preference-Optimized Multi-Task Vehicle Routing Solver with Preference Gating

**Dian Meng** [1 2]   **Yaoxin Wu** [3]   **Yaqing Hou** [1 2 *]   **Zhiguang Cao** [4]

## Abstract

Multi-task vehicle routing solvers via deep reinforcement learning have attracted broad attention and achieved significant progress in handling multiple constraints. However, existing neural solvers still face critical challenges, including insufficient representation, unstable training, and inefficient exploration in large combinatorial action spaces, which often prevents performance from meeting its full potential. To address these issues, we propose PoMtVRS (Preference-Optimized Multi-Task Vehicle Routing Solver with Preference Gating), a plug-and-play framework that jointly improves decoder representations and exploration efficiency through a synergistic combination of decoder-side augmentation and preference-driven optimization. Specifically, we introduce the preference optimization objective to learn relative comparisons among candidate solutions for different routing tasks, encouraging a higher generation probability of better solutions. Meanwhile, we design a preference-gated block that adaptively modulates decoder representations via sparse gated attention and nonlinear residual refinement. Extensive experiments demonstrate that PoMtVRS elevates state-of-the-art unified neural VRP backbones, achieving leading performance in multi-task benchmarks and stronger generalization. The code is available at: `https://github.com/Regina921/PoMtVRS`.

---

[1]School of Computer Science and Technology, Dalian University of Technology (DUT) [2]Key Laboratory of Social Computing and Cognitive Intelligence (DUT), Ministry of Education, China [3]Department of Industrial Engineering and Innovation Sciences, Eindhoven University of Technology [4]School of Computing and Information Systems, Singapore Management University. Correspondence to: Yaqing Hou <houyq@dlut.edu.cn>.

*Proceedings of the 43^{rd} International Conference on Machine Learning*, Seoul, South Korea. PMLR 306, 2026. Copyright 2026 by the author(s).

## 1. Introduction

Vehicle Routing Problems (VRPs) are a class of classic combinatorial optimization problems(COPs) in operations research and computer science, widely recognized as NP-hard, with broad applications in transportation services (Ge et al., 2019), logistics (Konstantakopoulos et al., 2022), and intelligent manufacturing (Zhang et al., 2023). Traditional solvers mainly fall into exact methods and handcrafted heuristics. Exact algorithms (e.g., branch-and-bound/branch-and-cut) typically incur exponential computational cost as the problem size grows, which makes large-scale deployment expensive (Laporte & Nobert, 1983). Heuristics can produce near-optimal solutions within reasonable time, yet often rely heavily on domain expertise and extensive iterative search, which limits portability and efficiency under frequent constraint switching (Helsgaun, 2017; Vidal et al., 2012).

Recently, Neural Combinatorial Optimization (NCO) has emerged as a promising paradigm for VRPs (Bengio et al., 2021). By learning constructive or improvement-based neural heuristics, NCO solvers can generate solutions autoregressively and optimize policies end-to-end with trajectory sampling (e.g., REINFORCE (Williams, 1992)), achieving competitive solution quality and inference speed on specific VRP variants (Kwon et al., 2020; Drakulic et al., 2023; Luo et al., 2023; Meng et al., 2025a; Luo et al., 2025; Wu et al., 2026), together with some out-of-distribution generalization (Bi et al., 2022; Hou et al., 2023; Zheng et al., 2024; Zhou et al., 2023b). However, practical applications involve diverse node attributes and solution constraints. Under such variability, specialized neural solvers often require training separate models or extensive retuning for each variant, which leads to high maintenance cost and limited scalability.

Consequently, the community has begun to move toward unified multi-task VRP solvers that exploit shared structure across variants for knowledge transfer and combinatorial generalization, e.g., modeling variants as compositions of constraint attributes (Liu et al., 2024; Zhou et al., 2024), improving unified environments (Berto et al., 2025), and enhancing unified solvers through better architecture design, dynamic context understanding, and training recipes (Li et al., 2025; Pan et al., 2025b; Liu et al., 2025; Gui et al., 2026). Despite strong progress, two bottlenecks remain.

First, optimal decision rules can change drastically across constraint switches, while standard decoders largely rely on a fixed attention context, making it difficult to perform fine-grained, state-conditional modulation during decoding and often resulting in limited expressiveness and unstable training. Second, most neural solvers rely on reinforcement learning (RL) with absolute reward values, where reward scales vary across variants, and return differences shrink as policies improve, weakening the learning signal. Combined with cross-task gradient interference, this causes high variance, inefficient exploration, and slow convergence.

To address these issues, we propose PoMtVRS (Preference-Optimized Multi-Task Vehicle Routing Solver with Preference Gating), a plug-and-play framework that jointly improves decoder expressiveness and exploration efficiency via architectural augmentation and preference-driven optimization. First, to mitigate attention sink and unstable training under constraint switching, we insert a preference-gated block (PGB) after the multi-head attention (MHA) output. The block combines head-wise gating, state mapping, and nonlinear residual refinement, enabling constraint-conditioned modulation of attention pathways and contextual features for more robust decoding representations. Second, to overcome reward-scale mismatch and diminishing learning signals, we introduce preference optimization (PO) to learn relative preferences from pairwise comparisons between candidate solutions of the same instance. Based on the Bradley-Terry (BT) model of log-likelihood differences, we transform quantitative reward signals into qualitative preference signals, effectively improving exploration efficiency and solution quality. Finally, PoMtVRS can be seamlessly integrated into existing neural solvers and has demonstrated its efficiency and practicality on two representative neural methods. It achieved state-of-the-art performance on multi-task VRPs and exhibits excellent generalization capabilities across different problem scales. Our contributions are summarized as follows:

- We propose PoMtVRS, a plug-and-play preference-learning framework for multi-task VRPs, which jointly enhances representation capacity and exploration efficiency through cooperative decoder augmentation and preference-driven optimization.

- We design a preference-gated block that adaptively re-organizes decoder representations via head-wise gated attention modulation and nonlinear residual refinement. Coupled with the PO objective, we model pairwise preferences with the BT likelihood so that the two work in synergy to enable more stable optimization and more efficient exploration of the solution space.

- Extensive experiments show that PoMtVRS can be seamlessly integrated into state-of-the-art neural back-bone methods, achieving heading performance and exhibiting strong generalization capability.

## 2. Related Works

### 2.1. Neural combinatorial optimization for VRPs

NCO has become a prominent learning paradigm for solving VRPs, where deep policies are trained as generalizable heuristics. Existing methods are commonly categorized into construction-based solvers that generate solutions autoregressively and improvement-based solvers that iteratively refine an initial solution. Early work models routing as sequential decision making and trains autoregressive policies end-to-end with REINFORCE, enabling direct instance-to-route generation (Vinyals et al., 2015; Bello et al., 2017; Nazari et al., 2018). The Transformer-based attention model (AM) substantially improves solution quality and scalability and has become a dominant backbone for single-task neural solvers (Kool et al., 2019), while POMO further boosts performance by exploiting solution symmetries with multi-start sampling (Kwon et al., 2020). Subsequent studies enhance constructive frameworks and explore stronger inductive representations (Kwon et al., 2021; Li et al., 2021; Kim et al., 2022; Berto et al., 2023; Chen et al., 2023; Chalumeau et al., 2023; Drakulic et al., 2023; Luo et al., 2023; Hottung et al., 2024; Meng et al., 2025b). In parallel, improvement-based methods combine learned local operators with large-neighborhood search to achieve better solutions under larger inference budgets (Chen & Tian, 2019; Lu et al., 2020; Hottung & Tierney, 2020; d O Costa et al., 2020; Wu et al., 2021; Xin et al., 2021; Hudson et al., 2022; Zhou et al., 2023a; Ma et al., 2023). Nevertheless, most neural solvers are still trained and tuned for a single variant under a fixed distribution, which can be brittle under constraint switching or unified multi-variant settings.

### 2.2. Multi-Task Learning for VRPs

Learning unified VRP solvers by exploiting shared structures across variants has gained increasing attention. Wang & Yu (2025) propose a multi-armed bandit scheme for task scheduling and sampling allocation to improve training efficiency under a fixed budget. Lin et al. (2024) pretrain a backbone on a canonical VRP and adapt it to target variants via low-dimensional modules, enabling parameter-efficient transfer. Drakulic et al. (2025) trains a generalist CO agent with supervised learning to promote cross-task reuse, while Jiang et al. (2024) leverages LLMs to map instance descriptions into a unified embedding space for representation sharing. For unified solvers, Liu et al. (2024) model variants as compositions of primitive constraint attributes for zero-shot generalization, and Zhou et al. (2024) improves multi-variant performance with mixture-of-experts. Berto et al. (2025) further enhances training stability and

scalability via modern Transformer components and mixed-batch training. Li et al. (2025) explicitly encodes constraint prompts with a dual-attention design. Beyond multi-task, Goh et al. (2025) study the more realistic multi-task and multi-distribution setting with hierarchical inductive biases, and Pan et al. (2025b) reuse basic solvers through plug-and-play expert adapters with gated mixtures to mitigate model growth. Despite substantial progress, most unified solvers still rely on RL with absolute reward values, making training sensitive to diminishing learning signals and gradient interference, which degrades stability and sample efficiency.

## 2.3. Preference Optimization

PO has attracted increasing attention because it can align models with human preferences without explicitly training a reward model as required by conventional Reinforcement Learning from Human Feedback (RLHF). RLHF typically learns a reward model from ranked feedback and then fine-tunes the policy via RL (Stiennon et al., 2020; Ouyang et al., 2022). To simplify this pipeline, Rafailov et al. (2023) proposed Direct Preference Optimization (DPO), which bypasses reward modeling and directly optimizes the likelihood using preference pairs. This line of work has since been extended to improve preference modeling and objective stability, including interpreting DPO as implicit reward modeling (Dong et al., 2023), generalizing pairwise supervision to ranking-based feedback (Song et al., 2024), and developing simpler losses (Xu et al., 2023) and reference-free variants such as SimPO (Meng et al., 2024). A survey provides a more comprehensive overview of these advances (Xiao et al., 2024). PO has also been increasingly adopted in NCO. Pan et al. (2025a) introduced PO for COPs, and Liao et al. (2025) further improved training stability and effectiveness by proposing COP-tailored preference construction and objective-guided losses. Fan et al. (2026) proposed POCCO, extending PO to multi-objective COPs. However, most existing studies focus on single-problem settings, and systematic investigation of PO for VRPs, especially under multi-task solver regimes, remains limited.

## 3. Preliminaries

### 3.1. Problem Definition

A VRP instance of size $N$ on a complete graph $\mathcal{G} = \{V, E\}$, where $V = \{v_0, v_1, \ldots, v_N\}(|V| = N + 1)$ consists of a depot $v_0$ and $N$ customers $\{v_1, \ldots, v_N\}$. Each node $v_i \in V$ has 2D coordinates $(x_i, y_i)$, and each customer additionally carries constraint attributes $c_i$. The edge set is $E = \{e(v_i, v_j) \mid 0 \leq i \neq j \leq N\}$, and each edge induces a travel cost $e_{ij}$. Vehicles are homogeneous with capacity $Q$. A solution $\tau$ is a set of sub-routes. Each sub-route starts from the depot, visits a subset of customers, and satisfies feasibility: every customer is served exactly once and all

constraints hold throughout execution. We minimize the total Euclidean length $c(\tau)$: $\tau^* = \arg\min_\tau c(\tau)$, i.e., the minimum-cost feasible routing plan. Taking the capacitated vehicle routing problem (CVRP) as the base variant, combining the capacity constraint (C) with any subset of the four additional constraints (O)/(B)/(L)/(TW) yields 16 VRP variants. **Capacity constraint (C):** Each customer $v_i$ has a demand $q_i$, and the vehicle load must satisfy $q \leq Q$. **Open Routes (O):** A vehicle is not required to return to the depot after finishing its assigned customers. **Backhauls (B):** Linehaul customers satisfy demands $b_i > 0$, while backhaul customers satisfy $b_i < 0$. A sub-route may include both types, but all linehaul customers must be visited before any backhaul customer. **Duration Limit (L):** Each sub-route must satisfy an upper bound on its total travel duration (or length). **Time Window (TW):** Each customer $v_i$ has a service time window $[e_i, f_i]$ and a service duration $s_i$. The vehicle must arrive no later than $f_i$. If it arrives early, it must wait until $e_i$ starting service, ensuring the service start time lies within $[e_i, f_i]$ (see Appendix A for detailed definitions).

### 3.2. Transformer Decoder

Most neural VRP solvers follow a Transformer architecture, where the encoder maps an instance graph to joint node representations $\mathbf{H} = [\mathbf{h}_0, \mathbf{h}_1, \ldots, \mathbf{h}_N]^\top \in \mathbb{R}^{(N+1) \times d}$ with model dimension $d$. At the decoding step $t$, a context vector $h_t \in \mathbb{R}^d$ is formed from the current partial solution and is used to produce the probability of the next node.

**QKV Linear Projections.** Let $H$ is the number of attention heads and $d_h = d/H$ is the per-head dimension. For each head $h \in \{1, \ldots, H\}$, the decoder projects the context and node embeddings to queries, keys, and values:

$$q_t^{(h)} = h_t W_Q^{(h)}, K^{(h)} = \mathbf{H} W_K^{(h)}, V^{(h)} = \mathbf{H} W_V^{(h)}, \quad (1)$$

where $W_Q^{(h)}, W_K^{(h)}, W_V^{(h)}$ are learnable parameters.

**Scaled Dot-Product Attention (SDPA).** Each head computes compatibility scores via scaled dot products and applies a feasibility mask, followed by a softmax normalization. The head output $z_t^{(h)}$ is the weighted sum of values:

$$z_t^{(h)} = \text{softmax}\left(\frac{q_t^{(h)} K^{(h)\top}}{\sqrt{d_h}}\right) V^{(h)}, \quad (2)$$

where $z_t^{(h)} \in \mathbb{R}^{1 \times d_h}$, and softmax$(\cdot)$ ensures the attention weights are non-negative and sum to 1 across each row.

**Multi-Head Concatenation.** The $H$ head outputs are concatenated along the feature dimension:

$$M_t^H = \text{Concat}\left(z_t^{(1)}, \ldots, z_t^{(H)}\right). \quad (3)$$

The SDPA output is fused by the output projection $W_o$:

$$A_t = W_o M_t^H, \quad (4)$$

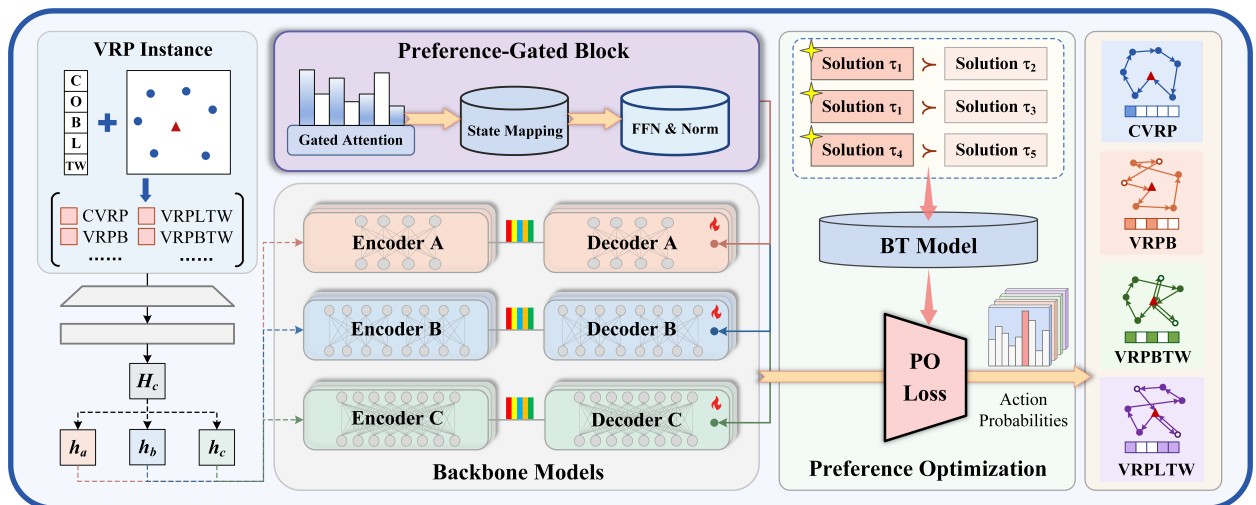

*Figure 1.* An overview of the proposed PoMtVRS. It integrates seamlessly into typical encoder-decoder frameworks: Under the unified VRP environment, PoMtVRS augments standard neural solvers by inserting a Preference-Gated Block (PGB) into the decoder, which enhances decoder representation and adaptively refines attention context. Training adopts preference optimization (PO), which replaces scalarized rewards by transforming intra-instance pairwise preferences into a Bradley-Terry (BT) likelihood and PO loss, yielding a stable reward signal to optimize the PL policy.

**Compatibility and Softmax.** Given the final vector $A_t$, we compute a score $\gamma_t$ using a masked single-head attention:

$$\gamma_t = \begin{cases} C \cdot \tanh\left(\frac{A_t K^{(h)}}{\sqrt{d_h}}\right), & \text{if unvisited,} \\ -\infty, & \text{otherwise,} \end{cases} \quad (5)$$

where $C$ is a scaling constant. The final selection probability $\rho_t$ for the step $t$ is computed via the softmax function:

$$\rho_t = \text{softmax}(\gamma_t), \quad (6)$$

At each decoding step, an action $\tau_t$ is sampled according to $\rho_t$. Repeating this process for $n$ steps yields the full solution.

### 3.3. Reinforcement Learning for Solving VRPs

A common NCO paradigm models VRP construction as a Markov decision process (MDP) and uses an attention-based policy $p_\theta$ (Kool et al., 2019). For instance $\mathcal{G}$, the decoder selects a feasible action $\tau_t$ at each step $t$ given state $s_t$ (defined by $\mathcal{G}$ and the partial solution). A complete solution is $\tau = (\tau_1, \ldots, \tau_T)$ with probability factorized as:

$$p_\theta(\tau \mid \mathcal{G}) = \prod_{t=1}^{T} p_\theta(\tau_t \mid s_t, \mathcal{G}), \quad (7)$$

where $p_\theta(\tau_t \mid s_t, \mathcal{G})$ is the conditional probability of choosing action $\tau_t$ at state $s_t$. After constructing $\tau$, the environment returns a scalar reward $R(\tau, \mathcal{G})$. Over an instance distribution $P(\mathcal{G})$, policy gradient training maximizes the expected reward as follows:

$$\mathcal{L}_{\text{RL}}(\theta) = \mathbb{E}_{\mathcal{G} \sim P(\mathcal{G})} \mathbb{E}_{\tau \sim p_\theta(\cdot \mid \mathcal{G})} \left[ R(\tau, \mathcal{G}) \right]. \quad (8)$$

The policy network is typically optimized using the REIN-FORCE, which maximizes the expected reward $\mathcal{L}_{\text{RL}}(\theta)$ via the following gradient estimator:

$$\nabla_\theta \mathcal{L}_{\text{RL}}(\theta) = \mathbb{E}_{p_\theta(\tau \mid \mathcal{G})} \left[ \left( R(\tau, \mathcal{G}) - b(\mathcal{G}) \right) \nabla_\theta \log p_\theta(\tau \mid \mathcal{G}) \right], \quad (9)$$

where $b(\mathcal{G})$ is a baseline that depends only on the instance $\mathcal{G}$ and reduces gradient variance. Despite its simplicity and lack of optimal supervision, REINFORCE depends on absolute rewards $R(\tau, \mathcal{G})$. In multi-task VRPs, variant-dependent reward scales and noise often induce high-variance gradients and baseline sensitivity, while shrinking within-instance reward gaps weaken the learning signal as training progresses. To address this, we replace absolute rewards with relative preferences via PO, yielding more stable training for multi-task VRP solvers.

## 4. Method

### 4.1. Overview

We propose PoMtVRS, a preference-driven framework in a unified VRP environment. Following the standard Transformer paradigm, an encoder produces node embeddings and attention caches, and an autoregressive decoder constructs feasible routes via masked decoding. PoMtVRS targets two major issues: (i) constraint switches alter decision rules, while fixed attention context limits state-conditional modulation and destabilizes training. (ii) RL with scalarized rewards suffers from reward scale mismatch and shrinking return gaps, amplifying gradient interference and hurting exploration. To address them, PoMtVRS inserts a lightweight

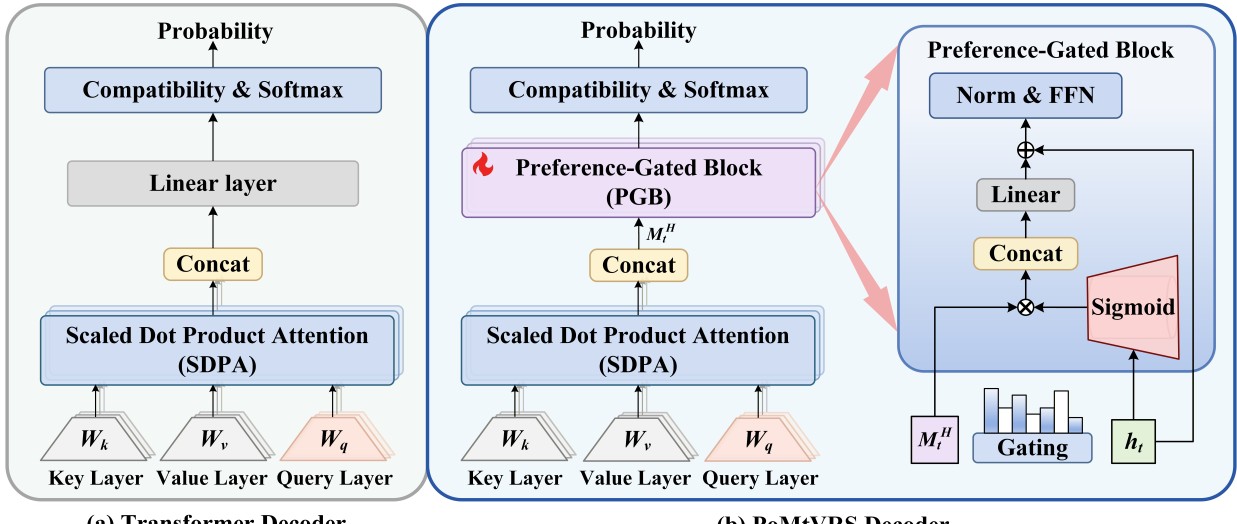

*Figure 2.* Decoder comparison between the backbone and PoMtVRS. (a) The standard decoder obtains a context representation via Scaled Dot Product Attention (SDPA) and directly applies a linear compatibility scoring layer followed by Softmax to produce action probabilities. (b) PoMtVRS inserts the Preference-Gated Block (PGB) between the attention output and the scoring layer to perform state-conditioned reorganization of the context vector. PGB combines head-wise gating, state mapping and nonlinear residual refinement using normalization and feed-forward. The refined context is then fed into the compatibility layer and softmax to compute probabilities.

preference-gated block and replaces RL with PO.

The overview of PoMtVRS is illustrated in Figure 1. For each instance, we perform multi-start rollouts to obtain candidate solutions and their trajectory log-likelihoods. We insert a plug-and-play preference-gated block between the MHA output and the compatibility layer to refine decoder representations. The block applies head-wise gating, attribute mapping, and nonlinear residual refinement, which improves robustness under diverse constraints (Section 4.2).

For training, PO converts raw rewards into within-instance pairwise preferences, so the policy assigns higher probability to higher-quality solutions. Because it depends on relative ordering rather than absolute values, PO mitigates reward-scale mismatch across tasks and provides denser supervision within each instance, which improves optimization stability and sample efficiency (Section 4.3). Overall, PoMtVRS combines decoder-side preference modulation with preference-driven optimization to form a stable, plug-and-play framework for neural solvers.

### 4.2. Preference-Gated Block (PGB)

Neural solvers typically use a Transformer decoder. At each step, a query from the partial solution attends to node keys/values to form a context vector, which is mapped to the next-node probability. In multi-task VRPs, constraint switches can alter the optimal policy, so attention context alone may miss heterogeneous signals. We thus introduce a lightweight preference-gated block for task-adaptive modulation, inserted between the MHA output $M_t^H$ (Section 3.2)

and the compatibility. The block has three components.

**Head-wise Gating.** Inspired by gated attention (Qiu et al., 2026), we introduce head-wise gating in the decoder. Instead of gating SDPA outputs, we apply the gating mechanism after multi-head concatenation, preserving the original MHA and enabling plug-and-play modulation. Specifically, we reshape $M_t^H$ by heads and compute a sigmoid gate $g_t^{(h)}$ from $h_t$, which modulates each head independently:

$$\tilde{y}_t^{(h)} = \mathcal{F}(M_t^H) \odot \sigma(h_t W_g), \qquad (10)$$

where $\sigma$ is an activation function (e.g., sigmoid), $W_g$ refers to the learnable parameters of gate, $\mathcal{F}(\cdot)$ is an inverse-concatenation operator, and $\tilde{y}_t^{(h)}$ is the gated output. The gated multi-head output is then projected to the model space to obtain $y_t \in \mathbb{R}^d$:

$$y_t = \text{Concat}(\tilde{y}_t^{(1)}, \ldots, \tilde{y}_t^{(H)})W_o. \qquad (11)$$

**State Attribute Mapping.** To explicitly inject constraint-related signals, we map state attribute $s_t$ into the embedding space and fuse it with the current node embedding $c_t$, forming a state-enhanced residual branch:

$$a_t = c_t + W_a s_t, \qquad (12)$$

where $W_a$ is learnable parameters of $s_t$, $s_t$ denotes the dynamic state determined by $\mathcal{G}$ and partial solution $\tau_{<t}$. The vector $a_t$ acts as a state-conditioned anchor, ensuring that subsequent fusion aligns with context embedding and state attributes.

**Nonlinear Residual Refinement.** Finally, we fuse $y_t$ with $a_t$ through residual normalization and feed-forward transformation to stabilize training and enhance expressiveness:

$$\tilde{u}_t = \text{Norm}(a_t + y_t), \tag{13}$$

$$u_t = \text{Norm}(\tilde{u}_t + \text{FFN}(\tilde{u}_t)), \tag{14}$$

where $\text{Norm}(\cdot)$ denotes a normalization operator, $\text{FFN}(\cdot)$ can be implemented as a feed-forward or a SigLU/parallel gated MLP, and $u_t$ is the preference-enhanced decoding representation used for node scoring and the next node.

### 4.3. Preference Optimization for solving VRPs

To mitigate the high-variance gradients and inefficient exploration of REINFORCE in multi-task VRP, we adopt PO, replacing raw scalarized rewards with relative preferences. PO encourages the policy to assign higher probability to better routes by generating multiple candidates per instance, forming preference pairs, and optimizing a Bradley-Terry (BT) likelihood on log-likelihood differences.

**Generating preference pairs.** Each instance $\mathcal{G}$ produces $P$ candidate routes $\{\tau^{(p)}\}_{p=1}^{P}$, which $P$ is the number of starts. The policy is $\pi_\theta(\tau \mid \mathcal{G})$ with parameters $\theta$. For a route $\tau^{(p)} = (\tau_1^{(p)}, \ldots, \tau_T^{(p)})$, we define log-likelihood as:

$$\ell^{(p)} = \log \pi_\theta(\tau^{(p)} \mid \mathcal{G}) = \sum_{t=1}^{T} \log \pi_\theta(\tau_t^{(p)} \mid \tau_{<t}^{(p)}, \mathcal{G}), \tag{15}$$

where $\tau_t^{(p)}$ is the action at step $t$. Let $c(\mathcal{G}, \tau^{(p)})$ be the environment cost (e.g., total tour length). In our setting, The reward is the negative cost: $r^{(p)} = -c(\tau^{(p)}, \mathcal{G})$. For numerical stability, we center rewards within the same instance $\tilde{r}^{(p)} = r^{(p)} - \frac{1}{P}\sum_{q=1}^{P} r^{(q)}$, which preserves pairwise ordering. We then define pairwise preference labels:

$$y_{i,j} = \mathbb{I}\left[\tilde{r}^{(i)} > \tilde{r}^{(j)}\right], \tag{16}$$

where $\mathbb{I}[\cdot]$ is the indicator function. Here, $y_{i,j} \in \{0,1\}$, $y_{i,j} = 1$ means that route $\tau^{(i)}$ achieves a higher (centered) reward than $\tau^{(j)}$ on the same instance.

**Bradley-Terry preference likelihood.** The BT model assigns each candidate a scalar preference strength and converts strength differences into win probabilities. We instantiate the strength of route $\tau^{(p)}$ as a scaled log-likelihood:

$$u^{(p)} = \alpha \ell^{(p)}, \tag{17}$$

where $u^{(p)} \in \mathbb{R}$ is the preference strength and $\alpha > 0$ is a temperature/scaling coefficient. The BT probability that route $i$ is preferred over route $j$ is:

$$\Pr_\theta(i \succ j \mid \mathcal{G}) = \sigma(u^{(i)} - u^{(j)}) = \sigma\big(\alpha(\ell^{(i)} - \ell^{(j)})\big), \tag{18}$$

where $\sigma(\cdot)$ is the sigmoid function. Equivalently, we define the pairwise logit $l_{i,j}$:

$$l_{i,j} = \alpha\big(\ell^{(i)} - \ell^{(j)}\big), \tag{19}$$

which directly quantifies the relative preference induced by policy between two candidate routes for the same instance.

**Preference optimization objective.** We maximize the log-likelihood of all preference comparisons, equivalently minimizing its negative, yielding per-instance PO objective:

$$\mathcal{L}_{\text{PO}}(\theta, \mathcal{G}) = -\frac{1}{P^2}\sum_{i=1}^{P}\sum_{j=1}^{P} y_{i,j} \log \sigma\Big(\alpha\big(\ell^{(i)} - \ell^{(j)}\big)\Big), \tag{20}$$

where $P^2$ denotes mean normalization over all pairs. Note that $y_{i,i} = 0$ since $\tilde{r}^{(i)} > \tilde{r}^{(i)}$ never holds, so diagonal terms contribute nothing. For batch size $B$, the training loss is:

$$\mathcal{L}_{\text{PO}}(\theta) = \frac{1}{B}\sum_{b=1}^{B} \mathcal{L}_{\text{PO}}(\theta, \mathcal{G}_b), \tag{21}$$

where $\mathcal{G}_b$ is the $b$-th instance. It enforces that if $\tau^{(i)}$ is preferred to $\tau^{(j)}$ (i.e., $y_{i,j} = 1$), the policy assigns higher probability to $\tau^{(i)}$, promoted by enlarging $\ell^{(i)} - \ell^{(j)}$. Compared to REINFORCE updates $\mathbb{E}[(r-b)\nabla_\theta \log p_\theta]$, PO uses relative log-likelihood differences, reducing sensitivity to reward scaling and baseline design.

## 5. Experiments

**Baselines.** *Traditional Solvers:* We benchmark against two classical VRP solvers. PyVRP (Wouda et al., 2024), built on top of HGS-CVRP (Vidal, 2022), serves as a strong heuristic baseline, while Google OR-Tools (Perron & Furnon, 2023) provides a robust industrial reference. Both solvers run on a single CPU core per instance with standard time limits that scale with problem size (e.g., 10s for VRP50 and 20s for VRP100). We parallelize traditional solvers across 16 CPU cores as in (Zhou et al., 2024). *Neural Solvers:* We compare against representative multi-task neural VRP solvers spanning unified training, expert specialization, and constraint-aware architectures. Specifically, we include MTPOMO (Liu et al., 2024), MVMoE (Zhou et al., 2024), RouteFinder (Berto et al., 2025) (RF-POMO, RF-MoE, RF-TE), CaDA (Li et al., 2025) and MoSES (Pan et al., 2025b). For each method, we train and evaluate the models under a unified hardware setup, following the same hyperparameter configurations and problem settings as in RouteFinder (Berto et al., 2025).

**PoMtVRS.** To validate plug-and-play compatibility, we instantiate PoMtVRS on two representative backbones: (i) MVMoE, a typical POMO-based method for multi-task

*Table 1.* Performance comparison on 1K test instances of 16 VRPs. Best learning-based results are highlighted. The lower, the better (↓).

| | Solver | N=50 Obj. | N=50 Gap | N=50 Time | N=100 Obj. | N=100 Gap | N=100 Time | | Solver | N=50 Obj. | N=50 Gap | N=50 Time | N=100 Obj. | N=100 Gap | N=100 Time |
|---|---|---|---|---|---|---|---|---|---|---|---|---|---|---|---|
| **CVRP** | HGS-PyVRP | 10.372 | * | 10.4m | 15.628 | * | 20.8m | **VRPTW** | HGS-PyVRP | 16.031 | * | 10.4m | 25.423 | * | 20.8m |
| | OR-Tools | 10.572 | 1.907% | 10.4m | 16.280 | 4.178% | 20.8m | | OR-Tools | 16.089 | 0.347% | 10.4m | 25.814 | 1.506% | 20.8m |
| | MTPOMO | 10.518 | 1.408% | 2s | 15.933 | 1.986% | 8s | | MTPOMO | 16.409 | 2.358% | 2s | 26.410 | 3.863% | 9s |
| | MVMoE | 10.508 | 1.316% | 3s | 15.912 | 1.848% | 11s | | MVMoE | 16.363 | 2.063% | 3s | 26.301 | 3.438% | 11s |
| | RF-POMO | 10.508 | 1.315% | 2s | 15.908 | 1.830% | 8s | | RF-POMO | 16.366 | 2.089% | 2s | 26.335 | 3.570% | 9s |
| | RF-MoE | 10.499 | 1.228% | 3s | 15.877 | 1.624% | 11s | | RF-MoE | 16.390 | 2.239% | 3s | 26.319 | 3.506% | 12s |
| | RF-TE | 10.504 | 1.276% | 2s | 15.857 | 1.507% | 8s | | RF-TE | 16.363 | 2.069% | 2s | 26.234 | 3.177% | 8s |
| | CaDA | 10.495 | 1.186% | 3s | 15.874 | 1.601% | 13s | | CaDA | 16.276 | 1.522% | 3s | 26.083 | 2.579% | 13s |
| | MoSES(CaDA) | 10.462 | 0.873% | 7s | 15.833 | 1.354% | 24s | | MoSES(CaDA) | 16.262 | 1.435% | 7s | 26.032 | 2.383% | 25s |
| | PoMtVRS(MVMoE) | 10.471 | **0.958%** | 4s | 15.819 | **1.256%** | 17s | | PoMtVRS(MVMoE) | 16.335 | **1.888%** | 4s | 26.245 | **3.213%** | 20s |
| | PoMtVRS(CaDA) | 10.459 | **0.843%** | 4s | 15.768 | **0.926%** | 17s | | PoMtVRS(CaDA) | 16.236 | **1.272%** | 4s | 25.976 | **2.151%** | 20s |
| **OVRP** | HGS-PyVRP | 6.507 | * | 10.4m | 9.725 | * | 20.8m | **VRPL** | HGS-PyVRP | 10.587 | * | 10.4m | 15.766 | * | 20.8m |
| | OR-Tools | 6.553 | 0.686% | 10.4m | 9.995 | 2.732% | 20.8m | | OR-Tools | 10.570 | 2.343% | 10.4m | 16.466 | 5.302% | 20.8m |
| | MTPOMO | 6.718 | 3.211% | 2s | 10.210 | 4.959% | 8s | | MTPOMO | 10.775 | 1.732% | 2s | 16.151 | 2.445% | 8s |
| | MVMoE | 6.710 | 3.086% | 3s | 10.206 | 4.921% | 11s | | MVMoE | 10.749 | 1.499% | 3s | 16.110 | 2.186% | 11s |
| | RF-POMO | 6.698 | 2.906% | 2s | 10.181 | 4.671% | 8s | | RF-POMO | 10.751 | 1.525% | 2s | 16.106 | 2.166% | 8s |
| | RF-MoE | 6.697 | 2.879% | 3s | 10.139 | 4.238% | 11s | | RF-MoE | 10.737 | 1.388% | 3s | 16.070 | 1.937% | 11s |
| | RF-TE | 6.684 | 2.693% | 2s | 10.121 | 4.060% | 8s | | RF-TE | 10.748 | 1.499% | 2s | 16.051 | 1.829% | 8s |
| | CaDA | 6.670 | 2.467% | 3s | 10.127 | 4.103% | 13s | | CaDA | 10.730 | 1.330% | 3s | 16.061 | 1.875% | 13s |
| | MoSES(CaDA) | 6.629 | 1.857% | 7s | 10.084 | 3.679% | 24s | | MoSES(CaDA) | 10.704 | 1.083% | 7s | 16.024 | 1.659% | 24s |
| | PoMtVRS(MVMoE) | 6.670 | **2.474%** | 4s | 10.084 | **3.674%** | 17s | | PoMtVRS(MVMoE) | 10.712 | **1.151%** | 4s | 16.013 | **1.571%** | 17s |
| | PoMtVRS(CaDA) | 6.640 | **2.010%** | 4s | 10.034 | **3.155%** | 17s | | PoMtVRS(CaDA) | 10.698 | **1.028%** | 4s | 15.951 | **1.188%** | 17s |
| **VRPB** | HGS-PyVRP | 9.687 | * | 10.4m | 14.377 | * | 20.8m | **OVRPTW** | HGS-PyVRP | 10.510 | * | 10.4m | 16.926 | * | 20.8m |
| | OR-Tools | 9.802 | 1.159% | 10.4m | 14.933 | 3.853% | 20.8m | | OR-Tools | 10.519 | 0.078% | 10.4m | 17.027 | 0.583% | 20.8m |
| | MTPOMO | 10.033 | 3.564% | 2s | 15.082 | 4.917% | 8s | | MTPOMO | 10.667 | 1.472% | 2s | 17.421 | 2.896% | 9s |
| | MVMoE | 10.008 | 3.303% | 3s | 15.036 | 4.600% | 11s | | MVMoE | 10.659 | 1.396% | 3s | 17.387 | 2.700% | 12s |
| | RF-POMO | 9.996 | 3.173% | 2s | 15.016 | 4.465% | 8s | | RF-POMO | 10.657 | 1.376% | 2s | 17.392 | 2.725% | 9s |
| | RF-MoE | 9.980 | 3.014% | 3s | 14.973 | 4.165% | 11s | | RF-MoE | 10.673 | 1.533% | 3s | 17.387 | 2.698% | 13s |
| | RF-TE | 9.978 | 2.996% | 2s | 14.942 | 3.950% | 8s | | RF-TE | 10.652 | 1.328% | 2s | 17.326 | 2.341% | 9s |
| | CaDA | 9.960 | 2.798% | 3s | 14.963 | 4.080% | 13s | | CaDA | 10.619 | 1.013% | 3s | 17.234 | 1.793% | 14s |
| | MoSES(CaDA) | 9.904 | 2.225% | 7s | 14.901 | 3.668% | 23s | | MoSES(CaDA) | 10.611 | 0.946% | 8s | 17.217 | 1.702% | 26s |
| | PoMtVRS(MVMoE) | 9.946 | **2.656%** | 4s | 14.913 | **3.739%** | 17s | | PoMtVRS(MVMoE) | 10.629 | **1.105%** | 4s | 17.301 | **2.188%** | 20s |
| | PoMtVRS(CaDA) | 9.899 | **2.171%** | 4s | 14.806 | **2.993%** | 17s | | PoMtVRS(CaDA) | 10.593 | **0.775%** | 4s | 17.175 | **1.443%** | 20s |
| **VRPBL** | HGS-PyVRP | 10.186 | * | 10.4m | 14.779 | * | 20.8m | **VRPBLTW** | HGS-PyVRP | 18.361 | * | 10.4m | 29.026 | * | 20.8m |
| | OR-Tools | 10.331 | 1.390% | 10.4m | 15.426 | 4.338% | 20.8m | | OR-Tools | 18.422 | 0.332% | 10.4m | 29.830 | 2.770% | 20.8m |
| | MTPOMO | 10.672 | 4.699% | 2s | 15.712 | 6.253% | 8s | | MTPOMO | 18.990 | 2.130% | 2s | 30.896 | 3.616% | 9s |
| | MVMoE | 10.607 | 4.072% | 3s | 15.628 | 5.692% | 11s | | MVMoE | 18.933 | 1.824% | 3s | 30.757 | 3.149% | 12s |
| | RF-POMO | 10.592 | 3.937% | 2s | 15.628 | 5.696% | 8s | | RF-POMO | 18.937 | 1.853% | 2s | 30.794 | 3.278% | 9s |
| | RF-MoE | 10.575 | 3.767% | 3s | 15.541 | 5.121% | 11s | | RF-MoE | 18.956 | 1.956% | 3s | 30.807 | 3.321% | 12s |
| | RF-TE | 10.578 | 3.798% | 2s | 15.528 | 5.038% | 8s | | RF-TE | 18.941 | 1.877% | 2s | 30.688 | 2.923% | 9s |
| | CaDA | 10.548 | 3.508% | 3s | 15.530 | 5.029% | 13s | | CaDA | 18.852 | 1.402% | 3s | 30.534 | 2.407% | 14s |
| | MoSES(CaDA) | 10.517 | 3.193% | 7s | 15.478 | 4.705% | 24s | | MoSES(CaDA) | 18.858 | 1.425% | 8s | 30.510 | 2.329% | 26s |
| | PoMtVRS(MVMoE) | 10.542 | **3.420%** | 4s | 15.493 | **4.783%** | 18s | | PoMtVRS(MVMoE) | 18.907 | **1.694%** | 4s | 30.711 | **2.997%** | 20s |
| | PoMtVRS(CaDA) | 10.494 | **2.966%** | 4s | 15.351 | **3.832%** | 18s | | PoMtVRS(CaDA) | 18.814 | **1.194%** | 4s | 30.429 | **2.048%** | 20s |
| **VRPBTW** | HGS-PyVRP | 18.292 | * | 10.4m | 29.467 | * | 20.8m | **VRPLTW** | HGS-PyVRP | 16.356 | * | 10.4m | 25.757 | * | 20.8m |
| | OR-Tools | 18.366 | 0.383% | 10.4m | 29.945 | 1.597% | 20.8m | | OR-Tools | 16.441 | 0.499% | 10.4m | 26.259 | 1.899% | 20.8m |
| | MTPOMO | 18.639 | 1.876% | 2s | 30.435 | 3.278% | 9s | | MTPOMO | 16.823 | 2.818% | 2s | 26.891 | 4.364% | 9s |
| | MVMoE | 18.594 | 1.624% | 3s | 30.306 | 2.836% | 12s | | MVMoE | 16.747 | 2.365% | 3s | 26.744 | 3.794% | 12s |
| | RF-POMO | 18.601 | 1.669% | 2s | 30.343 | 2.967% | 9s | | RF-POMO | 16.750 | 2.383% | 2s | 26.784 | 3.951% | 9s |
| | RF-MoE | 18.617 | 1.760% | 3s | 30.339 | 2.947% | 12s | | RF-MoE | 16.776 | 2.547% | 3s | 26.775 | 3.918% | 12s |
| | RF-TE | 18.600 | 1.675% | 2s | 30.240 | 2.618% | 9s | | RF-TE | 16.763 | 2.460% | 2s | 26.691 | 3.587% | 9s |
| | CaDA | 18.504 | 1.140% | 3s | 30.072 | 2.044% | 14s | | CaDA | 16.670 | 1.890% | 3s | 26.534 | 2.977% | 14s |
| | MoSES(CaDA) | 18.495 | 1.095% | 8s | 30.050 | 1.969% | 25s | | MoSES(CaDA) | 16.667 | 1.864% | 8s | 26.493 | 2.824% | 25s |
| | PoMtVRS(MVMoE) | 18.570 | **1.498%** | 4s | 30.263 | **2.689%** | 20s | | PoMtVRS(MVMoE) | 16.716 | **2.167%** | 4s | 26.708 | **3.647%** | 20s |
| | PoMtVRS(CaDA) | 18.478 | **0.995%** | 4s | 29.981 | **1.729%** | 20s | | PoMtVRS(CaDA) | 16.634 | **1.664%** | 4s | 26.433 | **2.581%** | 20s |
| **OVRPB** | HGS-PyVRP | 6.898 | * | 10.4m | 10.335 | * | 20.8m | **OVRPBL** | HGS-PyVRP | 6.899 | * | 10.4m | 10.335 | * | 20.8m |
| | OR-Tools | 6.928 | 0.412% | 10.4m | 10.577 | 2.315% | 20.8m | | OR-Tools | 6.927 | 0.386% | 10.4m | 10.582 | 2.363% | 20.8m |
| | MTPOMO | 7.108 | 3.004% | 2s | 10.878 | 5.224% | 8s | | MTPOMO | 7.112 | 3.056% | 2s | 10.883 | 5.272% | 8s |
| | MVMoE | 7.107 | 2.991% | 3s | 10.886 | 5.310% | 11s | | MVMoE | 7.120 | 3.166% | 3s | 10.904 | 5.472% | 11s |
| | RF-POMO | 7.086 | 2.689% | 2s | 10.836 | 4.823% | 8s | | RF-POMO | 7.087 | 2.695% | 2s | 10.837 | 4.835% | 8s |
| | RF-MoE | 7.080 | 2.613% | 3s | 10.806 | 4.526% | 11s | | RF-MoE | 7.083 | 2.635% | 3s | 10.807 | 4.540% | 11s |
| | RF-TE | 7.071 | 2.477% | 2s | 10.772 | 4.212% | 8s | | RF-TE | 7.075 | 2.515% | 2s | 10.779 | 4.268% | 8s |
| | CaDA | 7.047 | 2.129% | 3s | 10.761 | 4.083% | 13s | | CaDA | 7.049 | 2.146% | 3s | 10.762 | 4.093% | 13s |
| | MoSES(CaDA) | 7.034 | 1.942% | 7s | 10.726 | 3.765% | 24s | | MoSES(CaDA) | 7.036 | 1.964% | 7s | 10.724 | 3.743% | 24s |
| | PoMtVRS(MVMoE) | 7.055 | **2.243%** | 4s | 10.734 | **3.837%** | 17s | | PoMtVRS(MVMoE) | 7.060 | **2.309%** | 4s | 10.741 | **3.898%** | 18s |
| | PoMtVRS(CaDA) | 7.021 | **1.755%** | 4s | 10.657 | **3.093%** | 17s | | PoMtVRS(CaDA) | 7.022 | **1.755%** | 4s | 10.659 | **3.110%** | 18s |
| **OVRPBLTW** | HGS-PyVRP | 11.668 | * | 10.4m | 19.156 | * | 20.8m | **OVRPBTW** | HGS-PyVRP | 11.669 | * | 10.4m | 19.156 | * | 20.8m |
| | OR-Tools | 11.681 | 0.106% | 10.4m | 19.305 | 0.767% | 20.8m | | OR-Tools | 11.682 | 0.109% | 10.4m | 19.303 | 0.757% | 20.8m |
| | MTPOMO | 11.817 | 1.259% | 3s | 19.637 | 2.494% | 9s | | MTPOMO | 11.814 | 1.231% | 3s | 19.635 | 2.484% | 9s |
| | MVMoE | 11.808 | 1.181% | 4s | 19.607 | 2.334% | 12s | | MVMoE | 11.806 | 1.163% | 4s | 19.600 | 2.301% | 13s |
| | RF-POMO | 11.805 | 1.155% | 3s | 19.608 | 2.344% | 10s | | RF-POMO | 11.804 | 1.148% | 3s | 19.608 | 2.343% | 10s |
| | RF-MoE | 11.823 | 1.307% | 4s | 19.607 | 2.334% | 13s | | RF-MoE | 11.823 | 1.300% | 4s | 19.606 | 2.327% | 13s |
| | RF-TE | 11.804 | 1.147% | 2s | 19.551 | 2.045% | 9s | | RF-TE | 11.805 | 1.151% | 2s | 19.551 | 2.046% | 9s |
| | CaDA | 11.761 | 0.778% | 3s | 19.441 | 1.468% | 14s | | CaDA | 11.759 | 0.757% | 3s | 19.441 | 1.472% | 14s |
| | MoSES(CaDA) | 11.761 | 0.781% | 8s | 19.440 | 1.470% | 26s | | MoSES(CaDA) | 11.760 | 0.773% | 8s | 19.441 | 1.475% | 26s |
| | PoMtVRS(MVMoE) | 11.775 | **0.894%** | 5s | 19.525 | **1.904%** | 20s | | PoMtVRS(MVMoE) | 11.772 | **0.868%** | 5s | 19.520 | **1.875%** | 20s |
| | PoMtVRS(CaDA) | 11.744 | **0.637%** | 5s | 19.395 | **1.224%** | 20s | | PoMtVRS(CaDA) | 11.744 | **0.634%** | 5s | 19.394 | **1.222%** | 20s |
| **OVRPL** | HGS-PyVRP | 6.507 | * | 10.4m | 9.724 | * | 20.8m | **OVRPLTW** | HGS-PyVRP | 10.510 | * | 10.4m | 16.926 | * | 20.8m |
| | OR-Tools | 6.552 | 0.668% | 10.4m | 10.001 | 2.791% | 20.8m | | OR-Tools | 10.497 | 0.114% | 10.4m | 17.023 | 0.728% | 20.8m |
| | MTPOMO | 6.719 | 3.229% | 2s | 10.214 | 5.000% | 8s | | MTPOMO | 10.670 | 1.503% | 2s | 17.420 | 2.892% | 9s |
| | MVMoE | 6.717 | 3.186% | 3s | 10.217 | 5.028% | 11s | | MVMoE | 10.659 | 1.395% | 3s | 17.384 | 2.684% | 12s |
| | RF-POMO | 6.701 | 2.951% | 2s | 10.180 | 4.662% | 8s | | RF-POMO | 10.657 | 1.372% | 3s | 17.392 | 2.727% | 9s |
| | RF-MoE | 6.696 | 2.870% | 3s | 10.141 | 4.253% | 11s | | RF-MoE | 10.673 | 1.532% | 3s | 17.385 | 2.690% | 12s |
| | RF-TE | 6.685 | 2.713% | 2s | 10.121 | 4.054% | 8s | | RF-TE | 10.652 | 1.330% | 2s | 17.327 | 2.348% | 9s |
| | CaDA | 6.669 | 2.451% | 3s | 10.127 | 4.100% | 13s | | CaDA | 10.620 | 1.021% | 3s | 17.234 | 1.793% | 14s |
| | MoSES(CaDA) | 6.629 | 1.846% | 7s | 10.081 | 3.652% | 24s | | MoSES(CaDA) | 10.611 | 0.940% | 8s | 17.219 | 1.714% | 26s |
| | PoMtVRS(MVMoE) | 6.673 | **2.517%** | 4s | 10.090 | **3.735%** | 17s | | PoMtVRS(MVMoE) | 10.628 | **1.100%** | 4s | 17.305 | **2.214%** | 20s |
| | PoMtVRS(CaDA) | 6.639 | **2.004%** | 4s | 10.033 | **3.146%** | 17s | | PoMtVRS(CaDA) | 10.593 | **0.776%** | 4s | 17.174 | **1.440%** | 20s |

*Table 2.* Ablation and comparison studies on the proposed PoMtVRS across 16 VRPs.

| Avg.Gap | CaDA | w.o.block | w.o.Gate | w.o.Map | w.o.PO | w.o.RL-Reld | w.o.PO-Reld | PoMtVRS(CaDA) |
|---------|------|-----------|----------|---------|--------|-------------|-------------|---------------|
| VRP50 | 1.721% | 1.579% | 1.427% | 1.423% | 1.621% | 1.714% | 1.438% | **1.405%** |
| VRP100 | 2.844% | 2.529% | 2.241% | 2.323% | 2.453% | 2.694% | 2.529% | **2.205%** |

VRPs, and (ii) CaDA, a recent state-of-the-art approach built on a unified VRP environment. The resulting variants are denoted as PoMtVRS(MVMoE) and PoMtVRS(CaDA).

**Training.** Following prior protocols (Berto et al., 2025; Li et al., 2025), we train on $N \in \{50, 100\}$ for 300 epochs with 100,000 online-generated instances per epoch. We use the Adam optimizer (Kingma & Ba, 2015) with a learning rate of $3 \times 10^{-4}$, a weight decay of $1 \times 10^{-6}$, a batch size of 256, and decay the learning rate by 0.1 at epochs 270 and 295. The temperature parameter $\alpha$ is 0.03. We conduct experiments to examine the impact of temperature parameters on PoMtVRS's performance (Appendix C.2). MVMoE and CaDA are trained from scratch using their original architectures (Appendix B) and hyperparameters (Appendix C.3). All experiments run on an NVIDIA Tesla V100-32GB GPU. Further detailed experimental settings are provided in Appendix C.1.

**Evaluation.** We evaluate using the same test dataset as in (Li et al., 2025). For each VRP variant and each problem size, we use a standardized test set of 1K randomly generated instances. For all neural solvers, we apply greedy rollout strategy with $\times 8$ augmentation (Kool et al., 2019), and we report the best solution among the eight rollouts. We report the average objective, the optimality gap, and inference time. The gap is computed with respect to the best heuristic solver results (marked by "*" in Table 1).

### 5.1. Experimental Results

We conduct a comprehensive benchmark over all 16 VRP variants against state-of-the-art neural solvers, with the complete results summarized in Table 1. For each task block, the last two rows report PoMtVRS instantiated with MVMoE and CaDA, respectively. Across different variants and problem scales, both PoMtVRS instantiations consistently strengthen their corresponding backbones, while PoMtVRS(CaDA) achieves the best overall performance with smaller optimality gaps on most tasks.

Quantitatively, PoMtVRS(MVMoE) improves MVMoE on all 16 variants at VRP50/VRP100, reducing the average gap from 2.227%/3.680% to 1.809%/2.951% on VRP50/VRP100, respectively. More importantly, PoMtVRS(CaDA) yields larger improvements over a stronger backbone. Compared with CaDA, it reduces the average gap from 1.721% to 1.405% on VRP50 and from 2.844% to 2.205% on VRP100, with improvements ob-

served across all 16 tasks. In addition, PoMtVRS(CaDA) also outperforms the strong prior method MoSES(CaDA) in both solution quality and inference efficiency. Specifically, the average gap is further reduced from 1.515% to 1.405% on VRP50 and from 2.631% to 2.205% on VRP100, while maintaining lower inference time. This comparison shows that PoMtVRS does not rely on expensive search or heavy model augmentation to obtain better solutions. Instead, it introduces a lightweight architectural enhancement that improves the backbone solver with limited computational overhead. Overall, PoMtVRS(CaDA) achieves leading performance with a favorable balance between solution quality and efficiency, demonstrating the effectiveness of the proposed framework for unified multi-task VRP solving.

### 5.2. Ablation Study

Table 2 reports the ablation results of PoMtVRS and the comparison with the ReLD (Huang et al., 2025) architecture. We evaluate 6 variants: **w.o. block**, which removes the PGB block while keeping PO; **w.o. Gate**, which removes the gating mechanism in PGB while keeping PO; **w.o. Map**, which removes the state attribute mapping branch in PGB while keeping PO; **w.o. PO**, which keeps PGB but replaces PO with RL; and two ReLD-style controls, **w.o. RL-ReLD** and **w.o. PO-ReLD**, which remove PGB and adopt the ReLD configuration under the RL or PO setup, respectively.

Across both problem scales, removing PGB (w.o. block) leads to consistent performance degradation, confirming that the proposed block is important for enhancing decoder representations. Removing either the gating mechanism (w.o. Gate) or the state attribute mapping branch (w.o. Map) weakens performance, indicating that effective multi-task adaptation relies on both preference-conditioned decoder modulation and state-aware signals for gated representation refinement. Keeping PGB but replacing PO with RL (w.o. PO) also degrades performance and remains noticeably below the PoMtVRS, indicating that PGB is most effective when coupled with the PO. The contrast between w.o. block and w.o. PO further shows that neither PO nor PGB alone is sufficient to achieve the best performance. Instead, their joint use brings complementary benefits.

We further compare with ReLD to examine whether the gains come from generic architectural changes. Under the same ReLD configuration, w.o. PO-ReLD outperforms w.o. RL-ReLD, confirming the benefit of PO. However, it still remains inferior to PoMtVRS, indicating that the improve-

*Table 3.* Performance comparison of multi-task VRP solvers on CVRPLib datasets.

| Set-X | MTPOMO | MVMoE | RF-MoE | RF-POMO | RF-TE | CaDA | MoSES(CaDA) | PoMtVRS(CaDA) |
|---|---|---|---|---|---|---|---|---|
| Avg. Gap($N < 251$) | 6.566% | 5.829% | 6.754% | 5.905% | 5.061% | 4.772% | 4.724% | 4.561% |
| Avg. Gap($251 \leq N < 501$) | 11.529% | 10.616% | 9.217% | 8.399% | 8.107% | 8.889% | 7.948% | 7.865% |
| Avg. Gap($501 < N \leq 1001$) | 30.190% | 18.918% | 14.994% | 13.188% | 12.253% | 14.199% | 12.814% | 12.651% |
| Avg. Gap | 15.863% | 11.693% | 10.253% | 9.108% | 8.428% | 9.230% | 8.441% | **8.326%** |

*Table 4.* Zero-shot results on unseen constraints across 32 VRPs.

| | Aug.Gap | MTPOMO | MVMoE | RF-TE | CaDA | PoMtVRS(CaDA) | | Aug.Gap | MTPOMO | MVMoE | RF-TE | CaDA | PoMtVRS(CaDA) |
|---|---|---|---|---|---|---|---|---|---|---|---|---|---|
| VRP50 | MD | 43.953% | 43.290% | 38.205% | 37.571% | **34.807%** | VRP100 | MD | 45.438% | 45.762% | 39.854% | 43.652% | **39.812%** |
| | MB | 8.709% | 8.182% | 8.718% | 8.263% | **7.770%** | | MB | 12.296% | 11.738% | 12.168% | 11.947% | **10.936%** |
| | MDMB | 52.048% | 53.723% | 47.698% | 49.084% | **43.827%** | | MDMB | 55.570% | 57.370% | 52.849% | 55.501% | **51.281%** |

ments mainly arise from the targeted coupling between PO and PGB rather than arbitrary structural modifications. Overall, the ablation results show that PO and PGB are both essential, and their combination improves optimization stability, representation enhancement, and exploration efficiency. Detailed results are provided in Appendix C.4.

### 5.3. Generalization on Real-World Instances

To further evaluate PoMtVRS on real-world instances, we conduct experiments on the CVRPLIB benchmark. Following the standard protocol adopted by prior methods (Berto et al., 2025; Li et al., 2025), we use X-set instances (Uchoa et al., 2017) with more than 100 nodes, covering problem sizes from 101 to 1001. Table 3 reports the averaged generalization performance of PoMtVRS(CaDA) against seven representative neural solvers on CVRPLIB. For scale-wise analysis consistent with prior work, we partition the X-set into three subsets according to the instance scale: $N < 251$, $251 \leq N < 501$, and $501 < N \leq 1001$.

Overall, PoMtVRS(CaDA) consistently strengthens the CaDA backbone across all instance scales and achieves the heading performance on the full benchmark, which indicates better scalability and more robust generalization. In particular, the consistent gains on larger instances suggest better scalability and more robust generalization under realistic routing distributions. Detailed per-instance results are provided in Appendix C.5.

### 5.4. Generalization to Unseen Constraints

To assess zero-shot generalization beyond the training constraint space, we evaluate on VRP50 and VRP100 under three unseen settings and report the results in Table 4.

**Multi-Depot (MD):** vehicles depart from one of three depots and return to the same depot (Berto et al., 2025); we build 16 MD variants by adding MD to each base problem in Table 1. **Mixed Backhauls (MB):** we remove the precedence between linehaul and backhaul customers and allow arbitrary service order under capacity constraints. We form 8 MB variants by replacing backhaul with MB in backhaul-

related problems. We further evaluate their composition (**MDMB**) to induce a stronger shift from unseen constraint combinations, resulting in 8 additional variants. Across MD, MB, and MDMB, PoMtVRS(CaDA) delivers the best solution quality at both scales, consistently surpassing the CaDA backbone and other neural baselines, which confirms robust generalization to constraint shifts and their compositions.

### 5.5. Further Enhancement with Efficient Active Search

In addition to the main comparison, we further investigate whether PoMtVRS can benefit from test-time adaptation. Specifically, we integrate efficient active search (EAS) (Hottung et al., 2022) into PoMtVRS(CaDA) by optimizing a lightweight instance-specific residual module during inference and further adopt LoRA to reduce the search overhead. As reported in Appendix D, experiments on 1K test instances across all 16 VRP variants show that EAS consistently improves solution quality, while the LoRA-based variant preserves most of the gains with substantially lower search cost. Notably, PoMtVRS(CaDA) with EAS or EAS-LoRA continues to outperform the corresponding CaDA-based counterparts under the same search setting, demonstrating that the proposed architecture remains effective and complementary to test-time search.

## 6. Conclusion, Limitation, and Future Work

This paper proposes PoMtVRS, a plug-and-play preference learning framework for multi-task VRPs that improves decoder adaptation under frequent constraint switches and strengthens exploration beyond scalarized rewards. Extensive experiments show that PoMtVRS integrates seamlessly with mainstream backbones and achieves leading performance and generalization, while ablation studies also confirm the effectiveness of the preference-gated block and the PO algorithm working together. Limitations remain in handling more complex real-world constraint systems and larger problem scales. Future work will extend PoMtVRS to richer constraints and broader real-world distributions, and further improve efficiency for large-scale deployment.

## Acknowledgments

This work was supported in part by the National Natural Science Foundation of China under Grant 62372081, the Liaoning Provincial Central-Guided Local Sci-Tech Development Program under Grant 2025JH6/101000005, the Dalian Science and Technology Innovation Fund under Grant 2024JJ12GX020, the Liaoning Provincial Natural Science Foundation Program under Grant 2024-MSBA-05 and the 111 Project under Grant D23006. This work is also supported by National Natural Science Foundation of China under grant No. 52472461 from the Second Research Institute of Civil Aviation Administration of China.

## Impact Statement

This paper presents work whose goal is to advance the field of Machine Learning. There are many potential societal consequences of our work, none which we feel must be specifically highlighted here.

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

# A. Multi-Task VRPs Details

## A.1. VRP Variants

A VRP instance of size $N$ on a complete graph $\mathcal{G} = \{V, E\}$, where $V = \{v_0, v_1, \ldots, v_N\}(|V| = N + 1)$ consists of a depot $v_0$ and $N$ customers $\{v_1, \ldots, v_N\}$. Each node $v_i \in V$ has 2D coordinates $(x_i, y_i)$. Each customer additionally carries constraint attributes $c_i$ (e.g., demand/backhaul amount, time-window parameters). The edge set is $E = \{e(v_i, v_j) \mid 0 \leq i \neq j \leq N\}$, and each edge induces a travel cost $e_{ij}$ (distance or travel time). Vehicles are homogeneous with capacity $Q$. A solution $\tau$ is a set of sub-routes. Each sub-route departs from the depot, visits a subset of customers, and must satisfy feasibility: every customer is served exactly once, and all route-level constraints are respected throughout execution (e.g., cumulative demand never exceeds Q). We define the objective $c(\tau)$ as the total Euclidean length and seek $\tau^* = \arg\min_\tau c(\tau)$, i.e., the minimum-cost feasible routing plan.

We take CVRP as the base variant, characterized by the capacity constraint (C). Other fundamental VRP variants are obtained by adding extra constraints, e.g., OVRP by enabling open routes (O), VRPB by imposing route Backhauls (B), VRPL by imposing route duration limits (L), and VRPTW by imposing route Time Window (TW). We use five basic constraints to construct the task family. Combining any subset of the four additional constraints (O)/(B)/(L)/(TW) on top of (C) yields 16 VRP variants ($2^4$ possible combinations): CVRP, OVRP, VRPB, VRPL, VRPTW, OVRPTW, OVRPB, OVRPL, VRPBL, VRPBTW, VRPLTW, OVRPBL, OVRPBTW, OVRPLTW, VRPBLTW, OVRPBLTW.

**Capacity constraint (C)** [$q \in [0, Q]$]. Each customer $v_i$ has a delivery demand $q_i$ served by vehicles with fixed capacity $Q > 0$. The vehicle load must never exceed $Q$.

**Open Routes (O)** [$o \in \{0, 1\}$]. A vehicle is not required to return to the depot after finishing its assigned customers. A sub-route may terminate at the last visited customer.

**Backhauls (B)** [$b \in [0, Q]$]. We generalize demands via $b_i$, allowing both deliveries and pickups. Linehaul customers satisfy $b_i > 0$ (deliver from depot), while backhaul customers satisfy $b_i < 0$ (pickup to depot). A route may serve both types, but all linehaul customers must precede any backhaul customer to avoid reloading and capacity conflicts.

**Duration Limit (L)** [$l \in [0, L]$]. Each sub-route is constrained by an upper bound on total travel duration (or length), balancing workload across vehicles.

**Time Window (TW)** [$e, s, f \in [0, T]^3$]. Each customer $v_i$ is associated with a service time window $[e_i, f_i]$ and a service duration $s_i$. The vehicle must arrive no later than $f_i$. If it arrives early, it must wait until $e_i$ to start service, ensuring the service start time lies within $[e_i, f_i]$.

## A.2. VRP Instance Generation

This section specifies the instance generation pipeline of the unified VRP environment (Berto et al., 2025), covering node attributes and global constraint attributes. The environment naturally supports multi-depot routing, where $m$ denotes the

*Table 5.* 16 VRP variants with five constraints.

| | Capacity (C) | Open Route (O) | Backhaul (B) | Duration Limit (L) | Time Window (TW) |
|---|:---:|:---:|:---:|:---:|:---:|
| CVRP | ✓ | | | | |
| OVRP | ✓ | ✓ | | | |
| VRPB | ✓ | | ✓ | | |
| VRPL | ✓ | | | ✓ | |
| VRPTW | ✓ | | | | ✓ |
| OVRPTW | ✓ | ✓ | | | ✓ |
| OVRPB | ✓ | ✓ | ✓ | | |
| OVRPL | ✓ | ✓ | | ✓ | |
| VRPBL | ✓ | | ✓ | ✓ | |
| VRPBTW | ✓ | | ✓ | | ✓ |
| VRPLTW | ✓ | | | ✓ | ✓ |
| OVRPBL | ✓ | ✓ | ✓ | ✓ | |
| OVRPBTW | ✓ | ✓ | ✓ | | ✓ |
| OVRPLTW | ✓ | ✓ | | ✓ | ✓ |
| VRPBLTW | ✓ | | ✓ | ✓ | ✓ |
| OVRPBLTW | ✓ | ✓ | ✓ | ✓ | ✓ |

number of depots and $n$ the number of customers. For fair comparison with mainstream neural baselines, the main paper focuses on the single-depot setting ($m = 1$).

**Locations.** For each instance, we independently sample $m + n$ 2D coordinates from the unit square:

$$(x_i, y_i) \sim U(0, 1), \quad i \in \{0, \dots, m + n - 1\}, \tag{22}$$

where $\{(x_i, y_i)\}_{i=0}^{m-1}$ denote the depot coordinates, $\{(x_i, y_i)\}_{i=m}^{m+n-1}$ represent the coordinates of the customers. Distances are computed by Euclidean metrics, $d_{ij} = \|(x_i, y_i) - (x_j, y_j)\|_2$. The vehicle speed is set to 1.0, so travel time equals distance.

**Vehicle capacity (C).** We consider homogeneous vehicles and use a single capacity scalar $C$ shared by all routes within an instance. Following common practice in multi-task VRP benchmarks, $C$ is set as a deterministic function of $n$:

$$C = \begin{cases} 30 + \left\lfloor \dfrac{n}{5} \right\rfloor + \left\lfloor \dfrac{1000}{5} + \dfrac{n - 1000}{33.3} \right\rfloor, & n > 1000, \\ 30 + \left\lfloor \dfrac{n}{5} \right\rfloor, & 20 < n \leq 1000, \\ 30, & \text{otherwise.} \end{cases} \tag{23}$$

When needed for stable learning, demands are normalized by $C$ and the normalized capacity is set to 1.

**Open Route (O).** The open-route attribute is indicated by a binary variable $o \in \{0, 1\}$. When $o=1$, the route can terminate at the last served customer (open route), when $o=0$, the vehicle must return to a depot (closed route). The effective proportion of open route instances during training/evaluation is controlled by the sampling recipe, enabling consistent multi-task mixtures without changing the underlying instance format.

**Distance Limit (L).** The distance-limit constraint assigns a per-route upper bound $\rho$ such that each sub-route starting from a depot must have length at most $\rho$. To guarantee feasibility, $\rho$ is sampled from an interval determined by the farthest customers:

$$\rho \sim U\big(2\, d_{\max}, \rho_{\max}\big), \qquad d_{\max} = \begin{cases} \max_{i \in \mathcal{C}} d_{0i}, & m = 1, \\ \min_{j \in \mathcal{D}} \max_{i \in \mathcal{C}} d_{ji}, & m > 1, \end{cases} \tag{24}$$

where $\mathcal{D}$ and $\mathcal{C}$ denote depots and customers, respectively, and $\rho_{\max}$ is a predefined upper bound (e.g., 3.0). This strategy ensures round-trip reachability from at least one depot while allowing varying constraint tightness. If the distance limit is inactive, we set $\rho = +\infty$.

**Linehaul and Backhaul Demands (B).** For the depot nodes, both linehaul and backhaul demands are set to zero. For each customer $v_i$ ($i \geq m$), we first sample two nonnegative integers $\tilde{q}_i^{\mathrm{LH}}, \tilde{q}_i^{\mathrm{BH}} \sim \mathrm{Unif}\{1, 2, \dots, 9\}$, and then draw a Bernoulli indicator $z_i \sim \mathrm{Bernoulli}(p_{\mathrm{BH}})$ with $p_{\mathrm{BH}}=0.2$ to assign the customer type:

$$(q_i^{\mathrm{LH}}, q_i^{\mathrm{BH}}) = \begin{cases} (\tilde{q}_i^{\mathrm{LH}}, 0), & z_i = 0 \quad \text{(linehaul)}, \\ (0, \tilde{q}_i^{\mathrm{BH}}), & z_i = 1 \quad \text{(backhaul)}. \end{cases} \tag{25}$$

Backhaul demands are activated only for variants that include the backhaul constraint. Otherwise, we set $q_i^{\mathrm{BH}}=0$ for all customers and use $q_i^{\mathrm{LH}}$ as the standard demand.

**Time Windows (TW).** Time windows assign each customer $v_i$ ($i \geq m$) an interval $[e_i, l_i]$ and a service duration $s_i$. We set the vehicle speed to 1.0 so that travel times numerically match Euclidean distances. For the depot, we use $e_0 = 0$, $s_0 = 0$, and an overall horizon $l_0 = T$. We fix the depot time window to $[e_0, l_0] = [0, T]$ with $T = 4.6$ and set the depot service time to $s_0 = 0$. For each customer $i \in \mathcal{C}$, we sample

$$s_i \sim U(0.15, 0.18), \qquad \Delta t_i \sim U(0.18, 0.20), \tag{26}$$

where $\Delta t_i$ is the window length. Let $d_i = d_{0i}$ for $m=1$, and $d_i = \min_{j \in \mathcal{D}} d_{ji}$ for $m > 1$. To ensure that the simple tour $(0, i, 0)$ remains feasible, we compute an upper bound for the window start time:

$$u_i^{\mathrm{up}} = \frac{T - s_i - \Delta t_i}{d_i} - 1, \tag{27}$$

then draw $u_i \sim U(0, 1)$ and set the start times of the time windows $e_i$:

$$e_i = \left(1 + (u_i^{\mathrm{up}} - 1)u_i\right) d_i, \qquad l_i = e_i + \Delta t_i. \tag{28}$$

If the time-window constraint is inactive, we set $e_i = 0$, $l_i = +\infty$, and $s_i = 0$ for all customers. During decoding, feasibility is enforced via action masking (e.g., requiring predicted arrival times not exceeding $l_i$), and return-to-depot feasibility is additionally checked under the closed-route setting.

**Variant Composition.** The above attributes are composed modularly. By activating different subsets of constraint flags, the environment instantiates multiple VRP variants within a single unified multi-task setting for both training and evaluation.

### A.3. Feasible Action Space

To derive the feasible action space in the unified modular VRP environment, we construct a boolean action mask $m_t(i) \in \{0, 1\}$ at each decoding step $t$, where $m_t(i) = 1$ indicates that selecting node $v_i$ next is feasible. Let the vehicle be at node $v_{\tau_t}$, with current time $w_t^{\mathrm{cur}}$ and cumulative length of the current sub-route $l_t^{\mathrm{cur}}$. The Euclidean distance is denoted by $\mathrm{Dist}(\cdot, \cdot)$. The vehicle speed is fixed to $1.0$, and time windows are normalized accordingly, so the travel time is numerically identical to the Euclidean distance. A candidate action $i$ is feasible only if it passes the following checks.

**(1) Visit-once and depot-action constraints.** Each customer can be visited at most once, hence visited customers are always masked afterwards. To avoid degenerate self-loops, if the previous action is a depot, the next action cannot be a depot. Moreover, when the vehicle is currently at a depot while some customers remain unserved, depot actions are disallowed to prevent the decoder from stalling. Once all customers are served, depot selection is enabled to terminate decoding.

**(2) Time-window feasibility.** When the time-window constraint is active, each customer $v_i$ has a service window $[w_i^{\mathrm{beg}}, w_i^{\mathrm{end}}]$ and a service duration $w_i^{\mathrm{dur}}$. The arrival time at $v_i$ is $a_t(i) = w_t^{\mathrm{cur}} + \mathrm{Dist}(v_{\tau_t}, v_i)$, and the service start time is $s_t(i) = \max(a_t(i), w_i^{\mathrm{beg}})$. Action $i$ is feasible only if

$$s_t(i) + w_i^{\mathrm{dur}} \leq w_i^{\mathrm{end}}, \tag{29}$$

otherwise it is masked out.

**(3) Distance-limit feasibility.** When the distance-limit constraint is active, the current sub-route length cannot exceed $l^{\mathrm{dur}}$. Thus, action $i$ is feasible only if

$$l_t^{\mathrm{cur}} + \mathrm{Dist}(v_{\tau_t}, v_i) \leq l^{\mathrm{dur}}. \tag{30}$$

**(4) Return-to-depot feasibility for closed routes.** When the open-route constraint is inactive (i.e., routes must return to the depot), we additionally require that after visiting $v_i$ the vehicle can finish service and return to the starting depot of the current sub-route before the system end time $w_0^{\mathrm{end}}$. Denote this starting depot by $v_{d(t)}$ (for the single-depot case, $v_{d(t)} = v_0$). Then action $i$ must satisfy the return-time feasibility

$$\max\left(w_t^{\mathrm{cur}} + \mathrm{Dist}(v_{\tau_t}, v_i), w_i^{\mathrm{beg}}\right) + w_i^{\mathrm{dur}} + \mathrm{Dist}(v_i, v_{d(t)}) \leq w_0^{\mathrm{end}}. \tag{31}$$

If the distance-limit constraint is also active, we further enforce the return-distance feasibility

$$l_t^{\mathrm{cur}} + \mathrm{Dist}(v_{\tau_t}, v_i) + \mathrm{Dist}(v_i, v_{d(t)}) \leq l^{\mathrm{dur}}. \tag{32}$$

When the open-route constraint is active, the return feasibility checks above are removed, while the local TW and distance-limit checks remain enforced.

**(5) Capacity and linehaul-backhaul precedence.** When capacity is active, selecting $v_i$ requires that its demand does not exceed the remaining capacity. For variants with both linehaul and backhaul, we maintain separate remaining capacities $Q_t^{\mathrm{LH}}$ and $Q_t^{\mathrm{BH}}$, and check $q_i^{\mathrm{LH}} \leq Q_t^{\mathrm{LH}}$ or $q_i^{\mathrm{BH}} \leq Q_t^{\mathrm{BH}}$ depending on the customer type. In addition, we impose within-sub-route precedence: if there exist unserved linehaul customers in the current sub-route, all backhaul customers are masked. Once the sub-route enters the backhaul phase, remaining linehaul customers are disallowed within that sub-route, preventing LB precedence violations.

**(6) Multi-depot extension.** For the multi-depot setting ($m > 1$), the rules remain unchanged except that depot-related terms in the return feasibility are evaluated with respect to the starting depot $v_{d(t)}$ of the current sub-route. We also mask evidently invalid depot actions (e.g., depots that cannot serve any remaining customers) to ensure a non-stalling decoding process.

# B. Backbone Method Details

## B.1. MVMoE Backbone: Multi-Task VRP Solver with Mixture-of-Experts

MVMoE builds upon a classic Transformer construction policy as a shared backbone, enabling cross-task reuse through a unified feature space. It further increases model capacity by injecting Mixture-of-Experts (MoE) layers, while controlling the computational overhead via sparse and hierarchical gating.

**Mixture-of-Experts (MoE).** To enhance multi-task modeling, MVMoE augments the shared backbone with MoE layers. A typical MoE layer consists of a routing network and $m$ experts $\{E_j(\cdot)\}_{j=1}^m$. Given a single input $x$, the router produces a sparse gating vector $g(x) \in \mathbb{R}^m$ (e.g., via Top-$K$ selection), and the MoE output is computed as:

$$\text{MoE}(x) = \sum_{j=1}^m g_j(x)\, E_j(x), \tag{33}$$

where only a small subset of experts is activated per token. In MVMoE, MoE is used to replace the dense FFN in the encoder to encourage expert-wise specialization for graph representation learning. In addition, MVMoE parameterizes key decoder mappings (e.g., the output projection after MHA) with MoE to improve policy expressiveness under heterogeneous constraints. The router can be made task-aware by conditioning on a task embedding, enabling task-specific expert selection within a single shared model.

**Encoder with MoE.** MVMoE adopts a Transformer encoder to embed node features and model global interactions. Let $X_l \in \mathbb{R}^{n \times d}$ denote the token matrix at the $l$-th layer, where $n$ is the number of nodes and $d$ is the embedding dimension. Each layer follows a residual pre-norm formulation:

$$\tilde{X}_l = \text{IN}(X_l + \text{MHA}(X_l)), \tag{34}$$
$$X_{l+1} = \text{IN}(\tilde{X}_l + \text{MoE}(\tilde{X}_l)), \tag{35}$$

where $\text{IN}(\cdot)$ denotes instance normalization and $\text{MHA}(\cdot)$ is multi-head attention. Compared with a standard dense Transformer, the FFN in each encoder layer is replaced by an MoE module, thereby increasing capacity with limited additional computation due to sparse routing.

**Decoder with MoE.** The decoder constructs a solution in an autoregressive manner. At step $t$, it forms a context embedding $c_t$ by aggregating the depot embedding, the embedding of the last selected node, and the current resource state. Based on $c_t$, the decoder queries the encoded node embeddings $H = \{h_i\}_{i=1}^n$ and computes scores over candidate nodes. With an infeasibility mask applied, the next-node distribution is given by:

$$u_{t,i} = \text{Score}(c_t, h_i), \tag{36}$$
$$p_t = \text{softmax}(\text{Mask}(u_t)), \tag{37}$$

where $\text{Mask}(\cdot)$ sets infeasible logits to $-\infty$ before softmax. To further expand capacity, MVMoE replaces the dense output projection following multi-head attention with an MoE module. Concretely, letting $\{\text{head}_r\}_{r=1}^M$ be the attention heads, the projected query is computed as

$$q_t = \text{MoE}(\text{Concat}(\text{head}_1, \ldots, \text{head}_M)), \tag{38}$$

which provides conditional specialization in the decoder while maintaining tractable complexity through sparse routing.

**Gating mechanisms: base sparse gating and hierarchical gating.** For sparse routing, MVMoE adopts *input-choice* gating: each token is assigned to its top-$k$ experts according to gate scores. Given token matrix $X \in \mathbb{R}^{n \times d}$, gate logits are computed as $A = XW_G$, and each token selects the top-$k$ entries in its corresponding row of $A$, followed by a softmax over the selected experts to obtain sparse gates. To better trade off performance and efficiency in decoding, MVMoE further introduces *hierarchical gating*. A problem-level gate $G_1$ first decides whether to use a sparse MoE branch or a dense branch:

$$\bar{x} = \text{Pool}(X), \tag{39}$$
$$r = \text{softmax}(\bar{x}W_{G_1}), \tag{40}$$

where $\text{Pool}(\cdot)$ aggregates node tokens into a problem representation. If $G_1$ selects the sparse branch, a node-level gate $G_2$ routes tokens to experts as above. Otherwise, a dense transformation $D(X)$ is applied. This design reduces the average number of routed tokens and expert activations while preserving robustness across diverse and out-of-distribution instances.

**Training objective.** MVMoE is trained with RL to minimize the expected cost. The overall loss combines (i) a REINFORCE-style policy-gradient objective $\mathcal{L}_a$ and (ii) a load-balancing regularizer $\mathcal{L}_b$ to prevent expert collapse:

$$\mathcal{L} = \mathcal{L}_a + \alpha \mathcal{L}_b, \tag{41}$$

where $\alpha$ controls the strength of load balancing. This objective jointly promotes strong routing performance and effective utilization of multiple experts under sparse or hierarchical gating.

### B.2. CaDA Backbone: Cross-Problem Routing Solver with Constraint-Aware Dual-Attention

CaDA (Li et al., 2025) targets multi-task VRPs with an end-to-end encoder-decoder policy. The encoder maps an VRP instance to node representations, and the decoder constructs a feasible solution autoregressively by selecting the next node under constraint-induced feasibility masks. To better accommodate heterogeneous constraint combinations, CaDA injects a constraint-aware prompt into the encoder and employs a dual-attention encoder to capture both global interactions and salient sparse dependencies.

**Constraint prompt.** To explicitly encode the activated constraints, CaDA represents the constraint configuration by a multi-hot vector $V \in \mathbb{R}^5$ and transforms it via a multi-layer perceptron (MLP) into a prompt embedding $P^{(0)}$:

$$P^{(0)} = \text{LayerNorm}(VW_a + b_a)W_b + b_b, \tag{42}$$

where $W_a, b_a, W_b$ and $b_b$ are learnable parameters. The prompt is concatenated with node embeddings and propagated through the encoder, thereby injecting constraint information into representation learning.

**Dual-attention encoder.** CaDA stacks $L$ encoder layers with two parallel branches: a *global* branch and a *sparse* branch. The global branch uses standard MHA and a feed-forward layer (SwiGLU), together with RMSNorm and residual connections, to preserve full-graph information flow. The sparse branch follows a similar attention and FFN structure but replaces dense attention with Top-$k$ sparse attention. For each query, only the Top-$k$ attention scores are retained while the rest are masked before softmax, emphasizing highly related node pairs and suppressing redundant interactions. At the end of each layer, the two branches exchange information through linear projections and residual fusion:

$$H_g^{(i)} = \tilde{H}_g^{(i)} + (\tilde{H}_s^{(i)}W_s + b_s), \quad H_s^{(i)} = \tilde{H}_s^{(i)} + (\tilde{H}_g^{(i)}W_g + b_g), \tag{43}$$

where $\tilde{H}_g^{(i)}$ and $\tilde{H}_s^{(i)}$ denote the intermediate outputs of the global and sparse branches at layer $i$, and $H_g^{(i)}$ and $H_s^{(i)}$ are the fused representations. The final node embeddings are typically taken from the global branch and fed into the decoder.

**Autoregressive decoding under feasibility.** At the decoding step $t$, CaDA concatenates the embedding of the previously selected node $h_{\tau_t}^{(L)}$ with a 5-D state feature vector $\mathbf{s}_t$ and applies a linear projection to obtain the context embedding $H_c$:

$$H_c = [h_{\tau_t}^{(L)}, \mathbf{s}_t]W_t, \quad \mathbf{s}_t = \left[ c_t^{\text{LH}}, c_t^{\text{BH}}, t_t, \ell_t, o_t \right] \in \mathbb{R}^5, \tag{44}$$

where $c_t^{\text{LH}}$ and $c_t^{\text{BH}}$ denote the remaining capacities for linehaul/backhaul, $t_t$ is the current time, $\ell_t$ is the current route length, and $o_t$ indicates whether an open route exists. The decoder computes compatibility scores $u_i$ over the feasible set $\mathcal{I}_t$, masks infeasible actions with $-\infty$, and applies Softmax function to obtain the action probabilities $p_i$:

$$u_i = \begin{cases} \xi \cdot \tanh\left( \frac{q_c^\top h_i^{(L)}}{\sqrt{d_k}} \right), & i \in \mathcal{I}_t, \\ -\infty, & \text{otherwise}, \end{cases} \tag{45}$$

$$p_i = \text{Softmax}(u_i), \tag{46}$$

where $\xi$ is the logit clipping constant, $d_k$ is the feature dimension, and $q_c$ is the query vector derived from $H_c$.

## C. Experiment Details

### C.1. Experiment Setup

**Baselines.** *Traditional Solvers:* We benchmark against two widely used classical VRP solvers. PyVRP (Wouda et al., 2024), built on top of HGS-CVRP (Vidal, 2022), serves as a strong state-of-the-art heuristic baseline, while Google OR-Tools (Perron & Furnon, 2023) provides a robust industrial reference. Both solvers run on a single CPU core per instance

*Table 6.* Effect of the temperature parameter $\alpha$ on PoMtVRS across 16 VRPs.

| $\alpha$ | 0.001 | 0.003 | 0.005 | 0.01 | 0.03 | 0.05 | 0.08 | 0.1 | 1.0 |
|---|---|---|---|---|---|---|---|---|---|
| Avg.Obj | 11.489 | 11.495 | 11.493 | 11.491 | 11.482 | 11.483 | 39.180 | 39.180 | 20.780 |
| Avg.Gap(%) | 1.450 | 1.527 | 1.513 | 1.496 | **1.405** | 1.413 | 258.950 | 258.950 | 78.010 |

with standard time limits that scale with problem size (e.g., 10s for VRP50 and 20s for VRP100). We parallelize traditional solvers across 16 CPU cores as in (Zhou et al., 2024). *Neural Solvers:* We compare against representative multi-task neural VRP solvers spanning unified training, expert specialization, and constraint-aware architectures. Specifically, we include MTPOMO (Liu et al., 2024), MVMoE (Zhou et al., 2024), and the RouteFinder family (Berto et al., 2025), i.e., RF-POMO, RF-MoE, and RF-TE, which provide strong unified-model baselines under consistent training recipes. We further consider CaDA (Li et al., 2025) (a constraint-aware dual-attention mechanism) and MoSES (Pan et al., 2025b) (a mixture-of-specialized-experts solver for compositional VRP variants). For each method, we train and evaluate the models under a unified hardware setup, following the same hyperparameter configurations and problem settings as in RouteFinder (Berto et al., 2025).

**PoMtVRS Architecture.** To demonstrate the plug-and-play versatility of PoMtVRS, we integrate it into two representative backbones: (i) MVMoE, a typical POMO-based method for multi-task settings, and (ii) CaDA, a recent state-of-the-art approach built on a unified VRP environment. The resulting variants are denoted as PoMtVRS(MVMoE) and PoMtVRS(CaDA), respectively. For fair comparison, all models are trained under the same unified VRP environment with identical hyperparameter configurations, using the same datasets and the same number of training epochs.

**Training.** We train models on two instance sizes ($N \in \{50, 100\}$), following the standard protocol in prior work (Berto et al., 2025; Li et al., 2025; Pan et al., 2025b). Each model is trained for 300 epochs containing 100,000 VRP instances, where all attributes are generated online to ensure dynamic diversity. We use the Adam optimizer (Kingma & Ba, 2015) with a learning rate of $3 \times 10^{-4}$, weight decay of $1 \times 10^{-6}$, and batch size of 256. The learning rate is decayed by a factor of 0.1 at epochs 270 and 295. Training is performed over all available variants using the same data budget across all neural methods. For MVMoE and CaDA, we train separate models from scratch for VRP50 and VRP100 with their original model architectures (Appendix B) and the hyperparameters in Appendix C.3. All experiments are conducted on NVIDIA Tesla V100-32GB GPU.

**Evaluation.** We evaluate using the same test dataset as in (Berto et al., 2025; Li et al., 2025; Pan et al., 2025b). For each VRP variant and each problem size, we use a standardized test set of 1K randomly generated instances. For all neural solvers, we apply a greedy rollout strategy with $\times 8$ augmentation (Kool et al., 2019): each instance is transformed into eight equivalent augmented instances, and we report the best solution among the eight rollouts. We report the average objective and the optimality gap over the 1K test instances, as well as the inference time. The gap is computed with respect to the best heuristic solver results (marked by "*" in Table 1).

### C.2. Choice of Temperature Parameter $\alpha$

In the PO framework, the temperature parameter $\alpha$ is derived from the entropy-regularized objective and controls the exploration-exploitation trade-off. Specifically, a larger $\alpha$ encourages more exploration, whereas a smaller $\alpha$ places more emphasis on exploitation. Prior work (Pan et al., 2025a) provides a practical guideline that $\alpha$ should be chosen according to the solver's inherent exploration capacity and the problem family. For VRPs, effective values are typically found at the $10^{-2} \sim 10^{-3}$ scale. Following this guideline, we conducted a grid search over $\alpha$. Specifically, during model selection, we first considered the candidate set $\{0.005, 0.01, 0.05, 0.1, 0.5, 1.0, 2.0\}$. For PoMtVRS, we then performed a finer search around the empirically effective range and additionally evaluated $\{0.001, 0.003, 0.005, 0.01, 0.03, 0.05, 0.08, 0.1, 1\}$, with the results reported in Table 6. The results show that $\alpha = 0.03$ achieves the best performance in PoMtVRS. This suggests that the sensitivity of $\alpha$ is structured rather than arbitrary. Once the problem class and the backbone are fixed, the effective range becomes relatively clear.

### C.3. Hyperparameters

Table 7 summarizes the hyperparameter settings used in our main experiments for two representative multi-task backbones (MVMoE and CaDA) and their corresponding PoMtVRS instantiations. To ensure a fair comparison, all methods are trained and evaluated under the same training protocol, including identical optimizer and learning-rate scheduling, batch size,

*Table 7.* Experiment Hyperparameters.

| Hyperparameter | Value | Hyperparameter | Value | Hyperparameter | Value |
|---|---|---|---|---|---|
| **MVMoE Model** | | **CaDA Model** | | **Training** | |
| Embedding dimension $d_h$ | 128 | Embedding dimension $d_h$ | 128 | Batch size | 256 |
| Number of attention heads $M_h$ | 8 | Number of attention heads $M_h$ | 8 | Optimizer | AdamW |
| Number of encoder layers $L_e$ | 6 | Number of encoder layers $L_e$ | 6 | Learning rate (LR) | $3e^{-4}$ |
| Number of decoder layers $L_d$ | 1 | Number of decoder layers $L_d$ | 1 | Weight decay | $1e^{-6}$ |
| Tanh clipping $\xi$ | 10.0 | Tanh clipping $\xi$ | 10.0 | LR scheduler | MultiStepLR |
| Feedforward hidden dimension $d_f$ | 512 | Feedforward hidden dimension $d_f$ | 512 | LR milestones | [270, 295] |
| Routing level | node | Feedforward activation | SwiGLU | LR gamma | 0.1 |
| Routing method | Input choice | Feedforward structure | Gated MLP | Train data per epoch | 100,000 |
| Normalization | Instance | Normalization | RMSNorm | Training epochs | 300 |
| Number of experts | 4 | Sparse | Top-k | Number of tasks used for training | 16 |
| Top-k | 2 | Top-k | N/2 | Gradient clip value | 1.0 |

number of epochs and training samples per epoch, the number of tasks, and gradient clipping. The core model capacity is also matched across methods (e.g., embedding dimension, attention heads, encoder depth, and feed-forward width), while only method-specific components and mechanisms (e.g., MoE or gated modules with Top-k selection, normalization and sparsity choices) follow their original designs, so that the observed differences can be attributed to the proposed approach rather than disparate training or capacity settings.

### C.4. Ablation Study Details

In the main paper, we report the averaged ablation results over 16 VRP variants in Table 2 of Section 5.2. Table 8 further provides the detailed results on two problem scales (VRP50 and VRP100), including the objective values and average optimality gaps of each ablation variant and baseline across all tasks. These results offer a more fine-grained view of how PO and the PGB contribute to the final performance.

**Effectiveness of the Preference-gated Block.** In terms of the average optimality gap, the full PoMtVRS(CaDA) consistently achieves the best performance on both scales. It reduces the average gap of the CaDA backbone from 1.721% to 1.405% on VRP50 and from 2.844% to 2.205% on VRP100. When the entire PGB is removed while PO is retained (w.o. block), the average gap increases to 1.579% on VRP50 and 2.529% on VRP100, showing a clear degradation relative to the full model. This confirms that PGB is a key component for improving decoder representations under heterogeneous routing constraints. We further examine the internal components of PGB. Removing the gating mechanism (w.o. Gate) worsens the average gap to 1.437% on VRP50 and 2.245% on VRP100, indicating that preference-conditioned feature modulation is important for effective multi-task adaptation. Similarly, removing the state attribute mapping branch (w.o. Map) leads to average gaps of 1.424% and 2.323% on VRP50 and VRP100, respectively, which are also inferior to the full model. This suggests that state attribute mapping provides useful task- and state-aware signals for guiding the gated representation refinement, especially on larger instances where constraint interactions become more complex. Overall, the degradation caused by removing either the whole block or its internal components verifies that the improvement of PGB does not simply come from adding parameters, but from its structured integration of preference information, gating modulation, and state-aware mapping.

**Reinforcement Learning (RL) vs. Preference Optimization (PO).** With the same CaDA backbone, replacing standard RL with PO already brings consistent improvements. Specifically, comparing CaDA with w.o. block, the average gap decreases from 1.721% to 1.579% on VRP50 and from 2.844% to 2.529% on VRP100. This indicates that the relative preference signal provides a more stable and better-aligned learning objective than absolute-return optimization in multi-task VRP settings, where reward scales may vary across different variants. Meanwhile, keeping PGB but removing PO (w.o. PO) still improves over CaDA, achieving average gaps of 1.621% on VRP50 and 2.453% on VRP100. However, it remains clearly worse than the full PoMtVRS(CaDA), whose gaps are 1.405% and 2.205% on the two scales. These results show that PGB and PO are complementary: PGB enhances the decoder representation, while PO provides a more suitable optimization signal to fully exploit this representation enhancement.

**Preference-gated Block vs. ReLD-based Architecture.** To rule out the hypothesis that arbitrary architectural tweaks are sufficient, we compare against ReLD-style modifications. Under the ReLD configuration, introducing PO improves the average gap from 1.714% to 1.438% on VRP50 and from 2.694% to 2.529% on VRP100, confirming the effectiveness of PO beyond the proposed architecture. However, w.o. PO-ReLD is still inferior to PoMtVRS(CaDA), especially on VRP100

*Table 8.* Ablation and comparison studies on the proposed PoMtVRS across 16 VRPs.

| N=50 | CaDA | | w.o. block | | w.o. Gate | | w.o. Map | | w.o. PO | | w.o. RL-Reld | | w.o. PO-Reld | | PoMtVRS(CaDA) | |
|---|---|---|---|---|---|---|---|---|---|---|---|---|---|---|---|---|
| | Obj. | Gap | Obj. | Gap | Obj. | Gap | Obj. | Gap | Obj. | Gap | Obj. | Gap | Obj. | Gap | Obj. | Gap |
| CVRP | 10.495 | 1.186% | 10.479 | 1.032% | 10.462 | 0.875% | 10.458 | 0.833% | 10.477 | 1.012% | 10.491 | 1.142% | 10.461 | 0.861% | 10.459 | 0.843% |
| OVRP | 6.670 | 2.467% | 6.657 | 2.269% | 6.644 | 2.075% | 6.639 | 2.003% | 6.665 | 2.396% | 6.672 | 2.504% | 6.643 | 2.054% | 6.640 | 2.010% |
| VRPB | 9.960 | 2.798% | 9.934 | 2.529% | 9.902 | 2.206% | 9.902 | 2.205% | 9.933 | 2.525% | 9.958 | 2.783% | 9.901 | 2.196% | 9.899 | 2.171% |
| VRPBL | 10.548 | 3.508% | 10.529 | 3.308% | 10.496 | 2.985% | 10.490 | 2.937% | 10.521 | 3.236% | 10.547 | 3.488% | 10.495 | 2.969% | 10.494 | 2.966% |
| VRPBTW | 18.504 | 1.140% | 18.493 | 1.073% | 18.480 | 1.012% | 18.486 | 1.040% | 18.503 | 1.135% | 18.507 | 1.160% | 18.485 | 1.033% | 18.478 | 0.995% |
| OVRPB | 7.047 | 2.129% | 7.034 | 1.940% | 7.020 | 1.786% | 7.022 | 1.770% | 7.044 | 2.092% | 7.049 | 2.162% | 7.022 | 1.773% | 7.021 | 1.755% |
| OVRPBLTW | 11.761 | 0.778% | 11.751 | 0.696% | 11.745 | 0.646% | 11.746 | 0.656% | 11.757 | 0.747% | 11.758 | 0.757% | 11.745 | 0.644% | 11.744 | 0.637% |
| OVRPL | 6.669 | 2.451% | 6.656 | 2.261% | 6.644 | 2.072% | 6.641 | 2.024% | 6.666 | 2.406% | 6.674 | 2.532% | 6.642 | 2.051% | 6.639 | 2.004% |
| VRPTW | 16.276 | 1.522% | 16.261 | 1.428% | 16.249 | 1.349% | 16.246 | 1.338% | 16.268 | 1.470% | 16.278 | 1.533% | 16.249 | 1.346% | 16.236 | 1.272% |
| VRPL | 10.730 | 1.330% | 10.715 | 1.193% | 10.698 | 1.032% | 10.696 | 1.010% | 10.712 | 1.161% | 10.731 | 1.334% | 10.698 | 1.034% | 10.698 | 1.028% |
| OVRPTW | 10.619 | 1.013% | 10.604 | 0.876% | 10.595 | 0.786% | 10.596 | 0.802% | 10.608 | 0.909% | 10.613 | 0.957% | 10.597 | 0.806% | 10.593 | 0.775% |
| VRPBLTW | 18.852 | 1.402% | 18.843 | 1.342% | 18.821 | 1.228% | 18.828 | 1.267% | 18.842 | 1.346% | 18.843 | 1.356% | 18.837 | 1.313% | 18.814 | 1.194% |
| VRPLTW | 16.670 | 1.890% | 16.652 | 1.783% | 16.630 | 1.764% | 16.634 | 1.672% | 16.650 | 1.771% | 16.664 | 1.858% | 16.636 | 1.684% | 16.634 | 1.664% |
| OVRPBL | 7.049 | 2.146% | 7.037 | 1.962% | 7.022 | 1.763% | 7.023 | 1.770% | 7.044 | 2.078% | 7.050 | 2.154% | 7.024 | 1.784% | 7.022 | 1.755% |
| OVRPBTW | 11.759 | 0.757% | 11.752 | 0.696% | 11.745 | 0.643% | 11.746 | 0.653% | 11.757 | 0.742% | 11.759 | 0.758% | 11.746 | 0.649% | 11.744 | 0.634% |
| OVRPLTW | 10.620 | 1.021% | 10.604 | 0.876% | 10.593 | 0.775% | 10.596 | 0.796% | 10.608 | 0.908% | 10.612 | 0.951% | 10.597 | 0.807% | 10.593 | 0.776% |
| Avg. | 11.514 | **1.721%** | 11.500 | **1.579%** | 11.484 | **1.437%** | 11.484 | **1.423%** | 11.503 | **1.621%** | 11.513 | **1.714%** | 11.486 | **1.438%** | 11.482 | **1.405%** |

| N=100 | CaDA | | w.o. block | | w.o. Gate | | w.o. Map | | w.o. PO | | w.o. RL-Reld | | w.o. PO-Reld | | PoMtVRS(CaDA) | |
|---|---|---|---|---|---|---|---|---|---|---|---|---|---|---|---|---|
| | Obj. | Gap | Obj. | Gap | Obj. | Gap | Obj. | Gap | Obj. | Gap | Obj. | Gap | Obj. | Gap | Obj. | Gap |
| CVRP | 15.874 | 1.601% | 15.818 | 1.244% | 15.769 | 0.998% | 15.819 | 1.052% | 15.804 | 1.158% | 15.897 | 1.552% | 15.838 | 1.179% | 15.768 | 0.926% |
| OVRP | 10.127 | 4.103% | 10.076 | 3.581% | 10.043 | 3.257% | 10.066 | 3.319% | 10.057 | 3.388% | 10.133 | 4.000% | 10.083 | 3.493% | 10.034 | 3.155% |
| VRPB | 14.963 | 4.080% | 14.886 | 3.547% | 14.807 | 2.993% | 14.842 | 3.106% | 14.854 | 3.336% | 14.950 | 3.861% | 14.877 | 3.372% | 14.806 | 2.993% |
| VRPBL | 15.530 | 5.029% | 15.468 | 4.610% | 15.362 | 3.891% | 15.440 | 4.108% | 15.407 | 4.218% | 15.574 | 5.005% | 15.464 | 4.298% | 15.351 | 3.832% |
| VRPBTW | 30.072 | 2.044% | 30.029 | 1.889% | 29.990 | 1.760% | 30.054 | 1.808% | 30.042 | 1.945% | 30.072 | 1.868% | 30.107 | 1.991% | 29.981 | 1.729% |
| OVRPB | 10.761 | 4.083% | 10.700 | 3.501% | 10.660 | 3.121% | 10.680 | 3.227% | 10.686 | 3.371% | 10.748 | 3.875% | 10.704 | 3.457% | 10.657 | 3.093% |
| OVRPBLTW | 19.441 | 1.468% | 19.421 | 1.360% | 19.397 | 1.238% | 19.428 | 1.290% | 19.425 | 1.390% | 19.436 | 1.324% | 19.463 | 1.477% | 19.395 | 1.224% |
| OVRPL | 10.127 | 4.100% | 10.078 | 3.599% | 10.042 | 3.244% | 10.067 | 3.322% | 10.056 | 3.376% | 10.133 | 3.996% | 10.083 | 3.492% | 10.033 | 3.146% |
| VRPTW | 26.083 | 2.579% | 26.020 | 2.327% | 25.987 | 2.192% | 26.049 | 2.255% | 26.043 | 2.419% | 26.062 | 2.304% | 26.109 | 2.492% | 25.976 | 2.151% |
| VRPL | 16.061 | 1.875% | 16.009 | 1.551% | 15.954 | 1.200% | 16.025 | 1.313% | 15.988 | 1.418% | 16.111 | 1.852% | 16.056 | 1.504% | 15.951 | 1.188% |
| OVRPTW | 17.234 | 1.793% | 17.206 | 1.625% | 17.180 | 1.472% | 17.206 | 1.541% | 17.214 | 1.677% | 17.217 | 1.605% | 17.242 | 1.752% | 17.175 | 1.443% |
| VRPBLTW | 30.534 | 2.407% | 30.506 | 2.306% | 30.447 | 2.107% | 30.527 | 2.134% | 30.497 | 2.288% | 30.553 | 2.223% | 30.600 | 2.388% | 30.429 | 2.048% |
| VRPLTW | 26.534 | 2.977% | 26.502 | 2.845% | 26.439 | 2.600% | 26.514 | 2.669% | 26.494 | 2.824% | 26.563 | 2.854% | 26.569 | 2.877% | 26.433 | 2.581% |
| OVRPBL | 10.762 | 4.093% | 10.700 | 3.498% | 10.662 | 3.135% | 10.677 | 3.195% | 10.687 | 3.382% | 10.748 | 3.880% | 10.705 | 3.465% | 10.659 | 3.110% |
| OVRPBTW | 19.441 | 1.472% | 19.420 | 1.358% | 19.397 | 1.241% | 19.427 | 1.282% | 19.425 | 1.388% | 19.433 | 1.308% | 19.463 | 1.479% | 19.394 | 1.222% |
| OVRPLTW | 17.234 | 1.793% | 17.205 | 1.623% | 17.180 | 1.471% | 17.207 | 1.543% | 17.212 | 1.666% | 17.216 | 1.597% | 17.242 | 1.749% | 17.174 | 1.440% |
| Avg. | 18.173 | **2.844%** | 18.128 | **2.529%** | 18.082 | **2.245%** | 18.127 | **2.323%** | 18.118 | **2.453%** | 18.178 | **2.694%** | 18.163 | **2.529%** | 18.076 | **2.205%** |

where the gap remains 2.529% compared with 2.205% of the full model. This demonstrates that the performance gain is not merely due to adopting an alternative decoder modification but mainly stems from the targeted coupling between PO and the proposed PGB. Overall, the detailed ablation results confirm that PO, PGB, the gating mechanism, and the state attribute mapping branch all contribute to the final performance, and their joint design leads to more stable optimization, stronger representation refinement, and better generalization across diverse VRP variants.

## C.5. Generalization on Real-World Instances

We further evaluate PoMtVRS(CaDA) on the CVRPLIB instances from the X-set, following the standard protocol adopted by recent unified VRP solvers (e.g., MVMoE, RouteFinder, CaDA, and MoSES ()). This benchmark contains 100 instances ranging from 101 to 1,001 nodes. While Table 3 in the main text reports the size-binned average gaps, Table 9 provides instance-wise comparisons on every instance for all methods. For clarity, we partition the test set into three subsets by instance size, consistent with prior work: $N < 251$, $251 \le N < 501$, and $501 < N \le 1001$.

As summarized from Table 9, PoMtVRS(CaDA) consistently outperforms both the CaDA baseline and the strong competitor MoSES(CaDA) across all three subsets. Specifically, on $N < 251$, PoMtVRS(CaDA) reduces the average gap from 4.772% (CaDA) to 4.561%, also improving over 4.724% by MoSES(CaDA). On $251 \le N < 501$, PoMtVRS(CaDA) achieves 7.865%, substantially better than 8.889% (CaDA) and also better than 7.948% (MoSES (CaDA)). On the most challenging subset $501 < N \le 1001$, PoMtVRS(CaDA) reaches 12.651%, improving upon 14.199% (CaDA) and 12.814% (MoSES(CaDA)). Overall, across all 100 instances, PoMtVRS(CaDA) lowers the average gap from 9.230% (CaDA) to 8.326%, and surpasses 8.441% from MoSES(CaDA). These results show that, under identical evaluation budgets and inference settings, PoMtVRS(CaDA) achieves consistent improvements across problem scales, with larger gains on medium- and large-scale instances, indicating stronger robustness on realistic benchmarks and improved scalability. In addition, relative to other multi-task baselines, PoMtVRS(CaDA) attains a leading or highly competitive average gap, which supports its effectiveness on the real-world instances.

*Table 9.* Experimental results on CVRPLIB instances from the X set.

| Set-X | | MTPOMO | | | MVMoE | | | RF-MoE | | | RF-POMO | | | RF-TE | | | CaDA | | | MoSES(CaDA) | | | PoMtVRS(CaDA) | | |
|---|---|---|---|---|---|---|---|---|---|---|---|---|---|---|---|---|---|---|---|---|---|---|---|---|---|
| Instance | Opt. | Obj. | Gap | Time | Obj. | Gap | Time | Obj. | Gap | Time | Obj. | Gap | Time | Obj. | Gap | Time | Obj. | Gap | Time | Obj. | Gap | Time | Obj. | Gap | Time |
| X-n101-k25 | 27591 | 29470 | 6.810% | 0.4s | 29076 | 5.382% | 0.5s | 28934 | 4.868% | 0.5s | 29090 | 5.433% | 0.3s | 29048 | 5.281% | 0.4s | 28944 | 4.904% | 0.5s | 29110 | 5.505% | 0.8s | 28941 | 4.893% | 0.6s |
| X-n106-k14 | 26362 | 28029 | 6.323% | 0.3s | 27443 | 4.101% | 0.5s | 27292 | 3.528% | 0.6s | 27378 | 3.854% | 0.3s | 27159 | 3.023% | 0.4s | 27042 | 2.579% | 0.3s | 27051 | 2.614% | 0.8s | 26763 | 1.521% | 0.6s |
| X-n110-k13 | 14971 | 15100 | 0.862% | 0.3s | 15327 | 2.378% | 0.5s | 15260 | 1.930% | 0.5s | 15519 | 3.660% | 0.3s | 15314 | 2.291% | 0.3s | 15229 | 1.723% | 0.3s | 15332 | 2.411% | 0.8s | 15064 | 0.621% | 0.6s |
| X-n115-k10 | 12747 | 13433 | 5.382% | 0.4s | 13475 | 5.711% | 0.6s | 13638 | 6.990% | 0.5s | 13263 | 4.048% | 0.3s | 13060 | 2.455% | 0.4s | 13085 | 2.652% | 0.5s | 13085 | 2.652% | 0.8s | 12862 | 0.902% | 0.6s |
| X-n120-k6 | 13332 | 14051 | 5.393% | 0.4s | 13782 | 3.375% | 0.6s | 13908 | 4.320% | 0.6s | 14061 | 5.468% | 0.4s | 13765 | 3.248% | 0.4s | 13678 | 2.595% | 0.3s | 13619 | 2.153% | 0.8s | 13540 | 1.560% | 0.6s |
| X-n125-k30 | 55539 | 59015 | 6.259% | 0.4s | 58200 | 4.791% | 0.7s | 58587 | 5.488% | 0.6s | 58770 | 5.818% | 0.4s | 58570 | 5.457% | 0.4s | 57748 | 3.977% | 0.4s | 57620 | 3.747% | 1.0s | 57404 | 3.358% | 0.7s |
| X-n129-k18 | 28940 | 30176 | 4.271% | 0.4s | 29334 | 1.361% | 0.6s | 30039 | 3.798% | 0.6s | 29645 | 2.436% | 0.4s | 29500 | 1.786% | 0.5s | 29620 | 2.350% | 0.9s | 29630 | 2.384% | 1.0s | 29530 | 2.039% | 0.8s |
| X-n134-k13 | 10916 | 11707 | 7.246% | 0.4s | 11462 | 5.002% | 0.6s | 11439 | 4.791% | 0.6s | 11463 | 5.011% | 0.4s | 11624 | 6.486% | 0.4s | 11652 | 6.742% | 0.5s | 11573 | 6.019% | 1.0s | 11241 | 2.977% | 0.8s |
| X-n139-k10 | 13590 | 14058 | 3.444% | 0.4s | 14099 | 3.745% | 0.6s | 13917 | 2.406% | 0.6s | 13945 | 2.612% | 0.4s | 13812 | 1.634% | 0.4s | 13940 | 2.575% | 0.4s | 13877 | 2.112% | 0.9s | 13806 | 1.589% | 0.8s |
| X-n143-k7 | 15700 | 16626 | 5.898% | 0.4s | 16349 | 4.134% | 0.6s | 16655 | 6.083% | 0.6s | 16603 | 5.752% | 0.5s | 16257 | 3.548% | 0.4s | 16189 | 3.115% | 0.4s | 15980 | 1.783% | 0.9s | 16085 | 2.452% | 0.8s |
| X-n148-k46 | 43448 | 46648 | 7.365% | 0.5s | 45893 | 5.627% | 0.8s | 46542 | 7.121% | 0.8s | 46082 | 6.062% | 0.5s | 45026 | 3.632% | 0.6s | 45606 | 4.967% | 0.6s | 45600 | 4.953% | 1.0s | 45476 | 4.668% | 0.9s |
| X-n153-k22 | 21220 | 23514 | 10.811% | 0.5s | 23661 | 11.503% | 0.7s | 23906 | 12.658% | 0.7s | 22991 | 8.346% | 0.5s | 23478 | 10.641% | 0.6s | 23142 | 9.057% | 0.5s | 23310 | 9.849% | 1.0s | 22897 | 7.902% | 0.9s |
| X-n157-k13 | 16876 | 17922 | 6.198% | 0.5s | 17439 | 3.336% | 0.7s | 17801 | 5.481% | 0.8s | 17534 | 3.911% | 0.5s | 17315 | 2.601% | 0.5s | 17295 | 2.483% | 0.5s | 17317 | 2.613% | 1.0s | 17168 | 1.730% | 0.9s |
| X-n162-k11 | 14138 | 14616 | 3.381% | 0.5s | 14705 | 4.010% | 0.7s | 14524 | 2.730% | 0.7s | 14663 | 3.713% | 0.5s | 14664 | 3.720% | 0.5s | 14704 | 4.003% | 0.5s | 14677 | 3.812% | 1.0s | 14517 | 2.681% | 0.9s |
| X-n167-k10 | 20557 | 21662 | 5.375% | 0.5s | 21504 | 4.607% | 0.7s | 21481 | 4.495% | 0.7s | 21410 | 4.149% | 0.5s | 21425 | 4.222% | 0.5s | 21078 | 2.534% | 0.5s | 21384 | 4.023% | 1.0s | 21663 | 5.380% | 0.9s |
| X-n172-k51 | 45607 | 48960 | 7.352% | 0.6s | 47883 | 4.990% | 0.9s | 49726 | 9.032% | 1.0s | 48412 | 6.150% | 0.6s | 48162 | 5.602% | 0.7s | 48198 | 5.681% | 0.6s | 48145 | 5.565% | 1.0s | 47933 | 5.100% | 0.9s |
| X-n176-k26 | 47812 | 51989 | 8.736% | 0.5s | 52117 | 9.004% | 0.8s | 53626 | 12.160% | 0.9s | 52347 | 9.485% | 0.6s | 51501 | 7.716% | 0.6s | 51120 | 6.919% | 0.6s | 51612 | 7.948% | 1.0s | 52194 | 9.165% | 0.9s |
| X-n181-k23 | 25569 | 26572 | 3.923% | 0.6s | 26456 | 3.469% | 0.8s | 29154 | 14.021% | 0.9s | 26544 | 3.813% | 0.6s | 26097 | 2.065% | 0.6s | 26262 | 2.710% | 0.6s | 26143 | 2.245% | 1.0s | 26035 | 1.822% | 0.9s |
| X-n186-k15 | 24145 | 25236 | 4.519% | 0.5s | 25151 | 4.166% | 0.8s | 25140 | 4.121% | 0.9s | 25238 | 4.527% | 0.5s | 25153 | 4.175% | 0.6s | 25345 | 4.970% | 0.6s | 25246 | 4.560% | 1.0s | 25191 | 4.332% | 0.9s |
| X-n190-k8 | 16980 | 18369 | 8.180% | 0.5s | 19078 | 12.356% | 0.9s | 18217 | 7.285% | 0.9s | 18696 | 10.106% | 0.6s | 17871 | 5.247% | 0.6s | 17882 | 5.312% | 0.6s | 17569 | 3.469% | 1.0s | 17818 | 4.935% | 0.9s |
| X-n195-k51 | 44225 | 48310 | 9.237% | 0.7s | 46974 | 6.216% | 1.0s | 48965 | 10.718% | 1.0s | 47479 | 7.358% | 0.7s | 47396 | 7.170% | 0.7s | 46723 | 5.648% | 0.7s | 47358 | 7.358% | 2.0s | 47081 | 6.458% | 0.9s |
| X-n200-k36 | 58578 | 62041 | 5.912% | 0.6s | 61627 | 5.205% | 0.9s | 61696 | 5.323% | 1.0s | 61662 | 5.265% | 0.6s | 61139 | 4.372% | 0.6s | 61010 | 4.152% | 0.6s | 61089 | 4.287% | 2.0s | 61314 | 4.670% | 0.9s |
| X-n204-k19 | 19565 | 20652 | 5.556% | 0.6s | 20584 | 5.208% | 0.9s | 20466 | 4.605% | 1.0s | 20730 | 5.955% | 0.6s | 20531 | 4.937% | 0.6s | 20420 | 4.370% | 1.0s | 20397 | 4.252% | 2.0s | 20397 | 4.252% | 0.9s |
| X-n209-k16 | 30656 | 32333 | 5.470% | 0.6s | 32358 | 5.552% | 0.9s | 32145 | 4.857% | 0.9s | 32585 | 6.292% | 0.6s | 31876 | 3.980% | 0.6s | 32184 | 4.984% | 0.6s | 32053 | 4.557% | 2.0s | 32012 | 4.423% | 0.9s |
| X-n214-k11 | 10856 | 11699 | 7.765% | 0.6s | 11597 | 6.826% | 0.9s | 11534 | 6.245% | 0.9s | 11638 | 7.203% | 0.6s | 11668 | 7.480% | 0.6s | 11748 | 8.217% | 0.6s | 11716 | 7.922% | 1.0s | 11741 | 8.152% | 0.9s |
| X-n219-k73 | 117595 | 121980 | 3.729% | 0.8s | 124434 | 5.816% | 1.0s | 121627 | 3.429% | 1.0s | 123500 | 5.021% | 0.8s | 120344 | 2.338% | 0.9s | 120011 | 2.055% | 0.8s | 119710 | 1.799% | 2.0s | 120837 | 2.757% | 1.2s |
| X-n223-k34 | 40437 | 43381 | 7.280% | 0.7s | 42694 | 5.582% | 1.0s | 43097 | 6.578% | 1.0s | 42601 | 5.352% | 0.7s | 42251 | 4.486% | 0.7s | 42273 | 4.540% | 0.7s | 42128 | 4.182% | 2.0s | 42305 | 4.619% | 1.2s |
| X-n228-k23 | 25742 | 28523 | 10.803% | 0.7s | 28033 | 8.900% | 1.0s | 29590 | 14.948% | 1.0s | 28212 | 9.595% | 0.8s | 28699 | 11.487% | 0.8s | 27821 | 8.076% | 0.7s | 27724 | 7.699% | 2.0s | 27733 | 7.734% | 1.0s |
| X-n233-k16 | 19230 | 20644 | 7.353% | 0.7s | 20656 | 7.415% | 1.0s | 20507 | 6.641% | 1.0s | 20427 | 6.225% | 0.7s | 20761 | 7.962% | 0.7s | 20285 | 5.486% | 0.9s | 20623 | 7.244% | 2.0s | 20433 | 6.256% | 1.2s |
| X-n237-k14 | 27042 | 30047 | 11.112% | 0.7s | 29772 | 10.095% | 1.0s | 29514 | 9.141% | 1.0s | 30084 | 11.249% | 0.7s | 29595 | 9.441% | 0.7s | 29518 | 9.156% | 1.0s | 29883 | 10.505% | 2.0s | 29789 | 10.157% | 1.2s |
| X-n242-k48 | 82751 | 88179 | 6.559% | 0.8s | 87497 | 5.735% | 1.0s | 87832 | 6.140% | 1.0s | 87029 | 5.170% | 0.8s | 85704 | 3.569% | 0.9s | 85813 | 3.700% | 0.8s | 85643 | 3.495% | 2.0s | 86726 | 4.803% | 1.2s |
| X-n247-k50 | 37274 | 41610 | 11.633% | 0.8s | 40973 | 9.924% | 1.0s | 43153 | 15.772% | 1.0s | 41120 | 10.318% | 1.0s | 40642 | 9.036% | 0.9s | 39918 | 7.093% | 0.8s | 40736 | 9.288% | 2.0s | 41510 | 11.364% | 1.2s |
| Avg. Gap (N < 251) | | | 6.566% | | | 5.829% | | | 6.754% | | | 5.905% | | | 5.061% | | | 4.772% | | | 4.724% | | | **4.561%** | |
| X-n251-k28 | 38684 | 41211 | 6.532% | 0.7s | 41330 | 6.840% | 1.0s | 40691 | 5.188% | 1.0s | 40811 | 5.498% | 0.8s | 40127 | 3.730% | 0.8s | 40359 | 4.330% | 0.8s | 40290 | 4.152% | 2.0s | 40385 | 4.397% | 1.2s |
| X-n256-k16 | 18839 | 20400 | 8.286% | 0.7s | 20559 | 9.130% | 1.0s | 20015 | 6.242% | 1.0s | 20238 | 7.426% | 0.7s | 19994 | 6.131% | 0.8s | 20372 | 8.137% | 0.9s | 20068 | 6.524% | 2.0s | 19827 | 5.244% | 1.2s |
| X-n261-k13 | 26558 | 28741 | 8.220% | 0.7s | 28524 | 7.403% | 1.0s | 28203 | 6.194% | 1.0s | 28525 | 7.406% | 1.0s | 28510 | 7.350% | 0.9s | 28833 | 8.566% | 1.0s | 28577 | 7.602% | 2.0s | 29060 | 9.420% | 1.5s |
| X-n266-k58 | 75478 | 84617 | 12.108% | 0.9s | 82048 | 8.705% | 1.0s | 81135 | 7.495% | 1.0s | 81053 | 7.386% | 0.9s | 79832 | 5.769% | 0.9s | 80115 | 6.144% | 1.0s | 80036 | 6.039% | 2.0s | 79792 | 5.715% | 1.5s |
| X-n270-k35 | 35291 | 38146 | 8.090% | 1.0s | 38333 | 8.620% | 1.0s | 37401 | 5.979% | 1.0s | 38051 | 7.821% | 0.9s | 37382 | 5.925% | 1.0s | 37674 | 6.752% | 0.9s | 36923 | 4.624% | 2.0s | 37091 | 5.100% | 1.5s |
| X-n275-k28 | 21245 | 24688 | 16.206% | 0.8s | 25021 | 17.774% | 1.0s | 25241 | 18.809% | 1.0s | 24321 | 14.479% | 0.8s | 24187 | 13.843% | 0.9s | 24482 | 15.237% | 0.8s | 24312 | 14.436% | 2.0s | 24016 | 13.043% | 1.5s |
| X-n280-k17 | 33503 | 36677 | 9.474% | 0.8s | 36636 | 9.351% | 1.0s | 36538 | 9.059% | 1.0s | 35558 | 6.134% | 0.9s | 36653 | 9.402% | 0.9s | 36081 | 7.695% | 1.0s | 35494 | 5.943% | 2.0s | 36485 | 8.901% | 1.5s |
| X-n284-k15 | 20226 | 22474 | 11.114% | 0.8s | 22583 | 11.653% | 1.0s | 21857 | 8.064% | 1.0s | 21928 | 8.652% | 0.9s | 22154 | 9.532% | 0.8s | 22295 | 10.229% | 0.8s | 22071 | 9.122% | 2.0s | 22337 | 10.437% | 1.5s |
| X-n289-k60 | 95151 | 104159 | 9.467% | 0.9s | 102202 | 7.410% | 2.0s | 102267 | 7.479% | 2.0s | 101494 | 6.666% | 1.0s | 100418 | 5.535% | 1.0s | 99739 | 4.822% | 1.0s | 100080 | 5.180% | 2.0s | 100057 | 5.156% | 1.5s |
| X-n294-k50 | 47161 | 52769 | 11.891% | 0.9s | 50886 | 7.898% | 2.0s | 51924 | 10.099% | 1.0s | 51033 | 8.210% | 0.9s | 50637 | 7.370% | 1.0s | 49929 | 5.869% | 1.0s | 49877 | 5.759% | 2.0s | 50009 | 6.038% | 1.5s |
| X-n298-k31 | 34231 | 37652 | 9.994% | 0.9s | 37344 | 9.094% | 1.0s | 36808 | 7.528% | 1.0s | 36785 | 7.461% | 0.9s | 37163 | 8.565% | 0.9s | 36993 | 8.069% | 1.0s | 37068 | 8.288% | 2.0s | 36455 | 6.497% | 1.5s |
| X-n303-k21 | 21736 | 23556 | 8.373% | 0.9s | 23263 | 7.025% | 1.0s | 23027 | 5.939% | 1.0s | 23097 | 6.262% | 0.9s | 23442 | 7.849% | 0.9s | 23748 | 9.257% | 0.9s | 23548 | 8.336% | 2.0s | 23501 | 8.120% | 1.5s |
| X-n308-k13 | 25859 | 28736 | 11.126% | 0.9s | 28518 | 10.283% | 1.0s | 29079 | 12.452% | 1.0s | 28030 | 8.396% | 0.9s | 28326 | 9.540% | 0.9s | 28913 | 11.810% | 1.0s | 28440 | 9.981% | 2.0s | 28777 | 11.284% | 1.5s |
| X-n313-k71 | 94043 | 102253 | 8.730% | 1.0s | 100620 | 6.994% | 2.0s | 100714 | 7.094% | 2.0s | 100083 | 6.423% | 1.0s | 99564 | 5.871% | 1.0s | 98899 | 5.164% | 1.0s | 98931 | 5.198% | 2.0s | 99770 | 6.089% | 1.5s |
| X-n317-k53 | 78355 | 82587 | 5.401% | 1.0s | 83632 | 6.735% | 1.0s | 87360 | 11.493% | 2.0s | 81981 | 4.628% | 1.0s | 80542 | 2.791% | 1.0s | 80692 | 2.981% | 1.0s | 80472 | 2.702% | 2.0s | 80316 | 2.502% | 1.5s |
| X-n322-k28 | 29834 | 32593 | 9.248% | 1.0s | 33497 | 12.278% | 1.0s | 32143 | 7.739% | 1.0s | 32403 | 8.611% | 0.9s | 32658 | 9.466% | 1.0s | 33206 | 11.303% | 1.0s | 32541 | 9.074% | 2.0s | 32113 | 7.639% | 1.5s |
| X-n327-k20 | 27532 | 30646 | 11.310% | 1.0s | 30603 | 11.154% | 1.0s | 29649 | 7.689% | 1.0s | 29638 | 7.649% | 0.9s | 30953 | 12.425% | 1.0s | 30953 | 12.425% | 1.0s | 30089 | 9.287% | 2.0s | 30265 | 9.926% | 1.5s |
| X-n331-k15 | 31102 | 34734 | 11.678% | 0.9s | 33636 | 8.147% | 1.0s | 34431 | 10.703% | 1.0s | 33597 | 8.022% | 1.0s | 34048 | 9.472% | 1.0s | 34578 | 11.176% | 1.0s | 34014 | 9.363% | 2.0s | 34234 | 10.070% | 1.5s |
| X-n336-k84 | 139111 | 152846 | 9.873% | 1.0s | 149229 | 7.273% | 2.0s | 150468 | 8.164% | 2.0s | 147371 | 5.938% | 1.0s | 146620 | 5.398% | 1.0s | 146707 | 5.460% | 1.0s | 146465 | 5.286% | 2.0s | 147025 | 5.688% | 1.5s |
| X-n344-k43 | 42050 | 46619 | 10.866% | 1.0s | 46947 | 11.646% | 2.0s | 45143 | 7.356% | 2.0s | 46098 | 9.627% | 1.0s | 44914 | 6.811% | 1.0s | 45571 | 8.373% | 1.0s | 44746 | 6.411% | 2.0s | 44983 | 6.975% | 1.5s |
| X-n351-k40 | 25896 | 29243 | 12.925% | 1.0s | 28373 | 9.565% | 2.0s | 28728 | 10.936% | 2.0s | 28620 | 10.550% | 1.0s | 28236 | 9.036% | 1.0s | 28059 | 8.353% | 1.0s | 28130 | 8.627% | 2.0s | 27821 | 7.434% | 1.8s |
| X-n359-k29 | 51505 | 55778 | 8.296% | 1.0s | 56165 | 9.048% | 2.0s | 54690 | 6.184% | 2.0s | 55013 | 6.811% | 1.0s | 55122 | 7.023% | 1.0s | 55183 | 7.141% | 1.0s | 55158 | 7.093% | 2.0s | 55821 | 8.379% | 1.8s |
| X-n367-k17 | 22814 | 26132 | 14.544% | 1.0s | 25588 | 12.159% | 2.0s | 26470 | 16.025% | 2.0s | 25150 | 10.239% | 1.0s | 25522 | 11.870% | 1.0s | 25251 | 10.683% | 1.0s | 25489 | 11.725% | 2.0s | 25994 | 13.938% | 1.5s |
| X-n376-k94 | 147313 | 156857 | 6.190% | 1.0s | 156546 | 5.980% | 2.0s | 156077 | 5.662% | 2.0s | 158456 | 7.273% | 1.0s | 151975 | 2.885% | 1.0s | 151390 | 2.489% | 1.0s | 151614 | 2.641% | 2.0s | 152336 | 3.129% | 1.8s |
| X-n384-k52 | 65940 | 73705 | 11.776% | 1.0s | 73570 | 11.571% | 2.0s | 70853 | 7.451% | 2.0s | 71089 | 7.809% | 1.0s | 70411 | 6.871% | 1.0s | 70611 | 7.080% | 1.0s | 70479 | 6.884% | 2.0s | 70289 | 6.595% | 1.8s |
| X-n393-k38 | 38260 | 43533 | 13.782% | 1.0s | 44638 | 16.670% | 2.0s | 41843 | 9.365% | 2.0s | 42161 | 10.196% | 1.0s | 41552 | 8.604% | 1.0s | 42934 | 12.216% | 1.0s | 42192 | 10.277% | 3.0s | 41692 | 8.970% | 2.0s |
| X-n401-k29 | 66154 | 71565 | 8.179% | 1.0s | 71787 | 8.515% | 2.0s | 69432 | 5.046% | 2.0s | 70480 | 6.539% | 1.0s | 69430 | 4.952% | 1.0s | 69875 | 5.625% | 1.0s | 69991 | 5.800% | 3.0s | 69751 | 5.437% | 2.0s |
| X-n411-k19 | 19712 | 23869 | 21.089% | 1.0s | 23139 | 17.385% | 2.0s | 24162 | 22.575% | 2.0s | 22203 | 12.637% | 1.0s | 22849 | 15.914% | 1.0s | 23521 | 19.323% | 1.0s | 22768 | 15.503% | 2.0s | 23157 | 17.476% | 2.0s |
| X-n420-k130 | 107798 | 122761 | 13.881% | 2.0s | 116362 | 7.944% | 2.0s | 120841 | 12.099% | 2.0s | 118046 | 9.507% | 2.0s | 117418 | 8.924% | 2.0s | 115012 | 6.692% | 2.0s | 116853 | 8.400% | 3.0s | 116874 | 8.419% | 2.0s |
| X-n429-k61 | 65449 | 74261 | 13.464% | 1.0s | 74158 | 13.307% | 2.0s | 71017 | 8.507% | 2.0s | 71070 | 8.588% | 1.0s | 70164 | 7.204% | 2.0s | 70969 | 8.434% | 1.0s | 70617 | 7.896% | 3.0s | 69896 | 6.794% | 2.0s |
| X-n439-k37 | 36391 | 41165 | 13.119% | 1.0s | 42161 | 15.856% | 2.0s | 38998 | 7.164% | 2.0s | 39947 | 9.772% | 1.0s | 39752 | 9.236% | 1.0s | 41149 | 13.075% | 1.0s | 39697 | 9.085% | 3.0s | 39448 | 8.400% | 2.0s |
| X-n449-k29 | 55233 | 60162 | 8.924% | 1.0s | 60015 | 8.658% | 2.0s | 59919 | 8.484% | 2.0s | 59925 | 8.495% | 1.0s | 61144 | 10.702% | 1.0s | 60723 | 9.940% | 1.0s | 60723 | 9.940% | 3.0s | 61462 | 11.277% | 2.0s |
| X-n459-k26 | 24139 | 29543 | 22.387% | 1.0s | 29100 | 20.552% | 2.0s | 26995 | 11.831% | 2.0s | 27224 | 12.780% | 1.0s | 27347 | 13.290% | 2.0s | 28267 | 17.101% | 1.0s | 27510 | 13.965% | 3.0s | 27953 | 15.800% | 2.0s |
| X-n469-k138 | 221824 | 252031 | 13.618% | 2.0s | 245581 | 10.710% | 3.0s | 242533 | 9.336% | 3.0s | 242197 | 9.184% | 2.0s | 238904 | 7.700% | 2.0s | 237548 | 7.089% | 3.0s | 237001 | 6.842% | 3.0s | 237686 | 7.151% | 2.4s |
| X-n480-k70 | 89449 | 101314 | 13.265% | 2.0s | 100121 | 11.931% | 3.0s | 96042 | 7.371% | 3.0s | 96984 | 7.865% | 2.0s | 95032 | 6.242% | 2.0s | 95466 | 6.727% | 3.0s | 95211 | 6.442% | 3.0s | 94802 | 5.984% | 2.4s |
| X-n491-k59 | 66483 | 77536 | 16.625% | 2.0s | 75226 | 13.151% | 2.0s | 72443 | 8.963% | 2.0s | 72142 | 8.512% | 2.0s | 72618 | 9.228% | 2.0s | 71702 | 7.850% | 2.0s | 71730 | 7.891% | 3.0s | 71828 | 8.039% | 2.4s |
| Avg. Gap (251 ≤ N < 501) | | | 11.529% | | | 10.616% | | | 9.217% | | | 8.399% | | | 8.107% | | | 8.889% | | | 7.948% | | | **7.865%** | |
| X-n502-k39 | 69226 | 75711 | 9.368% | 2.0s | 77033 | 11.278% | 3.0s | 73557 | 6.256% | 3.0s | 74317 | 7.354% | 2.0s | 71908 | 3.874% | 2.0s | 72655 | 4.953% | 2.0s | 71682 | 3.548% | 3.0s | 71999 | 4.005% | 3.0s |
| X-n513-k21 | 24201 | 34910 | 44.250% | 2.0s | 32858 | 35.771% | 2.0s | 27867 | 15.148% | 2.0s | 29547 | 22.091% | 2.0s | 29422 | 21.573% | 2.0s | 29422 | 21.573% | 2.0s | 29130 | 20.404% | 3.0s | 28567 | 17.214% | 3.0s |
| X-n524-k153 | 154593 | 176491 | 14.165% | 2.0s | 171734 | 11.088% | 3.0s | 178794 | 15.655% | 3.0s | 172181 | 11.377% | 2.0s | 174150 | 12.651% | 2.0s | 168181 | 8.790% | 3.0s | 172580 | 11.635% | 3.0s | 172650 | 11.680% | 3.0s |
| X-n536-k96 | 94846 | 109897 | 15.869% | 2.0s | 106031 | 11.793% | 3.0s | 103862 | 9.506% | 3.0s | 103355 | 9.498% | 2.0s | 102355 | 7.919% | 2.0s | 102753 | 8.342% | 3.0s | 101712 | 7.239% | 4.0s | 102058 | 7.603% | 3.0s |
| X-n548-k50 | 86700 | 110984 | 28.009% | 2.0s | 104240 | 20.231% | 3.0s | 101294 | 16.833% | 3.0s | 101549 | 17.127% | 2.0s | 100850 | 16.321% | 2.0s | 102318 | 18.014% | 2.0s | 101918 | 17.552% | 3.0s | 100514 | 15.933% | 3.0s |
| X-n561-k42 | 42717 | 55936 | 30.946% | 2.0s | 53110 | 24.330% | 3.0s | 47544 | 11.300% | 3.0s | 47835 | 11.981% | 2.0s | 48282 | 15.020% | 2.0s | 50287 | 17.721% | 2.0s | 49363 | 15.558% | 4.0s | 48981 | 14.663% | 3.0s |
| X-n573-k30 | 50673 | 60884 | 20.151% | 2.0s | 62033 | 22.418% | 3.0s | 59670 | 17.755% | 3.0s | 57388 | 13.252% | 2.0s | 56048 | 10.607% | 2.0s | 55353 | 9.236% | 3.0s | 55058 | 8.654% | 4.0s | 58052 | 14.561% | 3.0s |
| X-n586-k159 | 190316 | 226245 | 18.879% | 3.0s | 212545 | 11.680% | 4.0s | 209373 | 10.013% | 4.0s | 210049 | 10.369% | 3.0s | 205654 | 8.059% | 3.0s | 204649 | 7.531% | 3.0s | 204848 | 7.636% | 5.0s | 205868 | 8.171% | 4.0s |
| X-n599-k92 | 108451 | 131035 | 20.824% | 3.0s | 126654 | 16.785% | 4.0s | 118761 | 9.507% | 3.0s | 120022 | 10.669% | 3.0s | 116840 | 7.735% | 3.0s | 117784 | 8.606% | 3.0s | 116938 | 7.826% | 4.0s | 116138 | 7.087% | 4.0s |
| X-n613-k62 | 59535 | 77555 | 30.268% | 3.0s | 73633 | 23.680% | 3.0s | 67477 | 13.340% | 3.0s | 66818 | 12.233% | 2.0s | 67523 | 13.454% | 3.0s | 69069 | 16.014% | 3.0s | 67730 | 13.765% | 4.0s | 67244 | 12.948% | 4.0s |
| X-n627-k43 | 62164 | 76776 | 23.506% | 3.0s | 70744 | 13.802% | 3.0s | 68747 | 10.590% | 4.0s | 69716 | 12.149% | 3.0s | 67523 | 8.621% | 3.0s | 69361 | 11.577% | 3.0s | 67896 | 9.221% | 4.0s | 69795 | 12.275% | 4.0s |
| X-n641-k35 | 63684 | 83138 | 30.548% | 3.0s | 71986 | 13.036% | 4.0s | 70691 | 11.003% | 4.0s | 71120 | 11.676% | 3.0s | 70631 | 10.900% | 3.0s | 73624 | 15.608% | 3.0s | 71974 | 13.017% | 4.0s | 72144 | 13.284% | 4.0s |
| X-n655-k131 | 106780 | 120771 | 13.103% | 3.0s | 118758 | 11.217% | 4.0s | 119665 | 12.067% | 4.0s | 117339 | 9.889% | 3.0s | 112289 | 5.159% | 3.0s | 110657 | 3.631% | 3.0s | 110267 | 3.266% | 5.0s | 111949 | 4.840% | 5.0s |
| X-n670-k130 | 146332 | 183183 | 25.183% | 3.0s | 168210 | 14.951% | 4.0s | 180539 | 23.376% | 4.0s | 166596 | 13.848% | 3.0s | 168829 | 15.374% | 3.0s | 161571 | 10.414% | 3.0s | 163051 | 11.425% | 5.0s | 166526 | 13.800% | 5.0s |
| X-n685-k75 | 68205 | 92701 | 35.915% | 3.0s | 82607 | 21.116% | 4.0s | 78039 | 14.418% | 4.0s | 77265 | 13.283% | 3.0s | 78473 | 15.055% | 3.0s | 77852 | 14.145% | 3.0s | 77132 | 13.088% | 5.0s | 78365 | 14.896% | 5.0s |
| X-n701-k44 | 81923 | 92723 | 13.183% | 3.0s | 89704 | 9.498% | 4.0s | 89743 | 9.546% | 4.0s | 90006 | 9.867% | 3.0s | 90580 | 10.567% | 3.0s | 92198 | 12.542% | 3.0s | 90703 | 10.717% | 5.0s | 92447 | 12.846% | 5.0s |
| X-n716-k35 | 43373 | 59383 | 36.912% | 3.0s | 52170 | 20.282% | 4.0s | 49166 | 13.356% | 4.0s | 49524 | 14.182% | 3.0s | 50505 | 16.674% | 3.0s | 49700 | 14.586% | 3.0s | 49405 | 13.907% | 5.0s | 49932 | 15.122% | 5.0s |
| X-n733-k159 | 136187 | 175848 | 29.122% | 4.0s | 156268 | 14.745% | 5.0s | 158156 | 16.131% | 5.0s | 154339 | 13.329% | 4.0s | 148581 | 9.101% | 4.0s | 146080 | 7.264% | 4.0s | 147334 | 8.185% | 5.0s | 148020 | 8.688% | 5.0s |
| X-n749-k98 | 77269 | 102208 | 32.276% | 4.0s | 92403 | 19.586% | 5.0s | 88483 | 14.513% | 5.0s | 87621 | 13.397% | 4.0s | 85046 | 10.065% | 4.0s | 85325 | 10.426% | 4.0s | 84712 | 9.633% | 6.0s | 84688 | 9.601% | 5.0s |
| X-n766-k71 | 114417 | 132968 | 16.213% | 4.0s | 130101 | 13.708% | 5.0s | 133549 | 16.721% | 6.0s | 126445 | 10.512% | 4.0s | 129866 | 13.502% | 4.0s | 127752 | 11.655% | 4.0s | 126387 | 10.462% | 6.0s | 134439 | 17.499% | 5.0s |
| X-n783-k48 | 72386 | 108577 | 49.997% | 4.0s | 96432 | 33.219% | 5.0s | 82299 | 13.695% | 5.0s | 82041 | 13.338% | 4.0s | 82839 | 14.441% | 4.0s | 87562 | 20.965% | 5.0s | 83864 | 15.857% | 6.0s | 83908 | 15.917% | 6.0s |
| X-n801-k40 | 73311 | 92125 | 25.663% | 4.0s | 87187 | 18.928% | 5.0s | 89100 | 21.537% | 6.0s | 88259 | 20.390% | 5.0s | 86121 | 17.474% | 4.0s | 94076 | 28.325% | 4.0s | 89478 | 22.053% | 6.0s | 86766 | 18.353% | 5.0s |
| X-n819-k171 | 158121 | 192102 | 21.491% | 5.0s | 178856 | 13.113% | 7.0s | 175286 | 10.856% | 6.0s | 177119 | 12.015% | 5.0s | 174446 | 10.324% | 5.0s | 172387 | 9.022% | 5.0s | 171676 | 8.573% | 7.0s | 170687 | 7.947% | 6.0s |
| X-n837-k142 | 193737 | 231002 | 19.235% | 5.0s | 230226 | 18.834% | 7.0s | 213765 | 10.338% | 7.0s | 215009 | 10.980% | 5.0s | 208669 | 7.707% | 5.0s | 209540 | 8.157% | 5.0s | 209031 | 7.894% | 7.0s | 207693 | 7.203% | 6.0s |
| X-n856-k95 | 88965 | 117243 | 31.786% | 5.0s | 105763 | 18.882% | 7.0s | 109164 | 22.704% | 7.0s | 99273 | 11.587% | 5.0s | 102312 | 15.003% | 5.0s | 100500 | 12.968% | 5.0s | 98914 | 11.183% | 7.0s | 97111 | 9.156% | 7.0s |
| X-n876-k59 | 99299 | 114212 | 15.018% | 5.0s | 114175 | 14.981% | 7.0s | 110476 | 11.256% | 7.0s | 112919 | 13.716% | 5.0s | 107477 | 8.236% | 5.0s | 109693 | 10.467% | 5.0s | 110843 | 11.625% | 7.0s | 108520 | 9.286% | 6.0s |
| X-n895-k37 | 53860 | 106062 | 96.922% | 6.0s | 70363 | 30.641% | 6.0s | 64648 | 20.030% | 6.0s | 64343 | 19.463% | 5.0s | 64225 | 19.244% | 5.0s | 73280 | 36.056% | 6.0s | 67830 | 25.938% | 8.0s | 65063 | 20.800% | 8.0s |
| X-n916-k207 | 329179 | 387367 | 17.677% | 6.0s | 374899 | 13.889% | 8.0s | 361709 | 9.882% | 9.0s | 360505 | 9.516% | 7.0s | 353039 | 7.248% | 7.0s | 351887 | 6.898% | 9.0s | 352488 | 7.081% | 10.0s | 352769 | 7.166% | 9.0s |
| X-n936-k151 | 132715 | 200816 | 51.314% | 7.0s | 161700 | 21.840% | 8.0s | 182393 | 37.432% | 8.0s | 162903 | 22.747% | 7.0s | 154847 | 16.676% | 7.0s | 155618 | 17.257% | 9.0s | 155618 | 17.257% | 9.0s | 162069 | 22.118% | 9.0s |
| X-n957-k87 | 85465 | 126220 | 47.686% | 7.0s | 124190 | 45.311% | 8.0s | 106292 | 24.369% | 8.0s | 104024 | 21.715% | 7.0s | 103089 | 20.621% | 7.0s | 108664 | 27.144% | 7.0s | 106903 | 25.084% | 9.0s | 102148 | 19.520% | 9.0s |
| X-n979-k58 | 118976 | 138987 | 16.819% | 7.0s | 132651 | 11.494% | 9.0s | 133186 | 11.944% | 8.0s | 133188 | 11.945% | 8.0s | 129633 | 8.957% | 7.0s | 133201 | 11.956% | 7.0s | 132728 | 11.559% | 9.0s | 131390 | 10.434% | 9.0s |
| X-n1001-k43 | 72355 | 132976 | 83.783% | 7.0s | 89175 | 23.246% | 9.0s | 85919 | 18.748% | 8.0s | 84377 | 16.616% | 8.0s | 85852 | 18.655% | 7.0s | 92974 | 28.497% | 7.0s | 93476 | 29.191% | 10.0s | 87059 | 20.322% | 9.0s |
| Avg. Gap (501 < N ≤ 1001) | | | 30.190% | | | 18.918% | | | 14.994% | | | 13.188% | | | 12.253% | | | 14.199% | | | 12.814% | | | **12.651%** | |
| Avg. Gap | | | 15.863% | | | 11.693% | | | 10.253% | | | 9.108% | | | 8.428% | | | 9.230% | | | 8.441% | | | **8.326%** | |

## D. Efficient Activate Search with LoRA for Performance

**Efficient active search (EAS) for better solution quality.**    To further improve the quality of solutions obtained during testing of PoMtVRS(CaDA) on synthetic datasets, this work adopts an efficient active search (added-layer update strategy). Following the EAS-Lay paradigm, we augment the pretrained decoder with a lightweight, instance-specific residual module while keeping all original model parameters fixed during search. The added module is optimized per batch through gradient-based updates, enabling fast instance-level adaptation under a fixed inference budget. For each test instance, an instance-specific residual transform is applied to the decoder multi-head attention output $M_t^H$:

$$\tilde{M}_t^H = M_t^H + \Big( \text{ReLU}(M_t^H W_1 + b_1)W_2 + b_2 \Big), \tag{47}$$

where $W_1, b_1, W_2$ and $b_2$ denote the trainable parameters during search. Given the input $h$, the added layer applies two linear transformations with a ReLU nonlinearity in between. $W_1, W_2$ and their corresponding biases $b_1, b_2$ are optimized via gradient descent throughout the search. To ensure that the added layer does not perturb the original model output at the beginning of search, $W_2$ and $b_2$ are initialized to zero.

**Low-rank adaptation (LoRA) for faster search.**    While EAS significantly improves solution quality through iterative online gradient updates, it also introduces considerable runtime overhead due to repeated backpropagation and updates of high-dimensional decoder projections. To mitigate the search cost while preserving the solution-quality improvements from EAS, we incorporate low-rank adaptation (LoRA), which shifts the online optimization from updating full high-dimensional weights to learning small, low-rank updates. This approach substantially reduces the number of trainable parameters and the optimization complexity, thereby lowering per-iteration update costs and improving efficiency. For a weight matrix $W$, LoRA parameterizes a low-rank update as:

$$W' = W + \frac{\alpha}{r} BA, \tag{48}$$

where $A \in \mathbb{R}^{r \times d_{\text{in}}}$ and $B \in \mathbb{R}^{d_{\text{out}} \times r}$ are trainable low-rank factors, while the original weight matrix $W$ is kept frozen. During the search, we update only the EAS-Lay parameters and the LoRA parameters $(A, B)$, while freezing all other weights. This reduces the parameter space for online adaptation, leading to effective improvements in fewer iterations, making early stopping more likely and significantly shortening the overall search time with comparable performance gains.

**Experiments and results.**    Table 10 reports results on 1K test instances across 16 VRP variants at two scales ($N = 50$ and $N = 100$). Without active search, PoMtVRS(CaDA) already yields a consistent improvement over the CaDA backbone across all variants and both scales, indicating that the preference-driven augmentation generalizes well under diverse constraint combinations. With EAS, both CaDA and PoMtVRS benefit from online refinement, but PoMtVRS(CaDA)-EAS remains consistently stronger and achieves the best overall solution quality among learning-based solvers, exhibiting smaller gaps on a broader range of variants and maintaining its advantage at the larger scale ($N = 100$), where search becomes more challenging. Finally, EAS-LoRA delivers a substantially better accuracy-efficiency balance. It preserves most of the gains of full EAS while significantly reducing the search time across variants, making it a practical default for deployment. In contrast, full EAS provides the strongest performance upper bound when runtime is less constrained.

*Table 10.* Experimenta results on 1K test instances across 16 VRPs.

| Task | Solver | N=50 Obj. | Gap | Time | N=100 Obj. | Gap | Time |
|---|---|---|---|---|---|---|---|
| CVRP | HGS-PyVRP | 10.372 | * | 10.4m | 15.628 | * | 20.8m |
| | OR-Tools | 10.572 | 1.907% | 10.4m | 16.280 | 4.178% | 20.8m |
| | CaDA | 10.495 | 1.186% | 3s | 15.874 | 1.601% | 13s |
| | CaDA-EAS | 10.404 | 0.307% | 8.3m | 15.701 | 0.488% | 1.09h |
| | PoMtVRS(CaDA) | 10.459 | 0.843% | 4s | 15.768 | 0.926% | 17s |
| | PoMtVRS(CaDA)-EAS | 10.387 | **0.144%** | 16.4m | 15.629 | **0.035%** | 1.32h |
| | PoMtVRS(CaDA)-EAS-LoRA | 10.400 | **0.273%** | 6.4m | 15.649 | **0.161%** | 1.03h |
| OVRP | HGS-PyVRP | 6.507 | * | 10.4m | 9.725 | * | 20.8m |
| | OR-Tools | 6.553 | 0.686% | 10.4m | 9.995 | 2.732% | 20.8m |
| | CaDA | 6.670 | 2.467% | 3s | 10.127 | 4.103% | 13s |
| | CaDA-EAS | 6.597 | 1.356% | 8.5m | 10.025 | 3.051% | 1.17h |
| | PoMtVRS(CaDA) | 6.640 | 2.010% | 4s | 10.034 | 3.155% | 17s |
| | PoMtVRS(CaDA)-EAS | 6.553 | **0.700%** | 17.3m | 9.929 | **2.075%** | 1.33h |
| | PoMtVRS(CaDA)-EAS-LoRA | 6.568 | **0.920%** | 7.5m | 9.958 | **2.374%** | 45.4m |
| VRPB | HGS-PyVRP | 9.687 | * | 10.4m | 14.377 | * | 20.8m |
| | OR-Tools | 9.802 | 1.159% | 10.4m | 14.933 | 3.853% | 20.8m |
| | CaDA | 9.960 | 2.798% | 3s | 14.963 | 4.080% | 13s |
| | CaDA-EAS | 9.843 | 1.591% | 8.3m | 14.817 | 3.055% | 1.14h |
| | PoMtVRS(CaDA) | 9.899 | 2.171% | 4s | 14.806 | 2.993% | 17s |
| | PoMtVRS(CaDA)-EAS | 9.787 | **1.010%** | 16.5m | 14.683 | **2.128%** | 1.30h |
| | PoMtVRS(CaDA)-EAS-LoRA | 9.804 | **1.193%** | 9.4m | 14.714 | **2.340%** | 43.3m |
| VRPBL | HGS-PyVRP | 10.186 | * | 10.4m | 14.779 | * | 20.8m |
| | OR-Tools | 10.331 | 1.390% | 10.4m | 15.426 | 4.338% | 20.8m |
| | CaDA | 10.548 | 3.508% | 3s | 15.530 | 5.029% | 13s |
| | CaDA-EAS | 10.379 | 1.859% | 8.4m | 15.321 | 3.630% | 1.15h |
| | PoMtVRS(CaDA) | 10.494 | 2.966% | 4s | 15.351 | 3.832% | 18s |
| | PoMtVRS(CaDA)-EAS | 10.320 | **1.290%** | 17.2m | 15.157 | **2.517%** | 1.31h |
| | PoMtVRS(CaDA)-EAS-LoRA | 10.354 | **1.614%** | 9.0m | 15.212 | **2.891%** | 51.4m |
| VRPBTW | HGS-PyVRP | 18.292 | * | 10.4m | 29.467 | * | 20.8m |
| | OR-Tools | 18.366 | 0.383% | 10.4m | 29.945 | 1.597% | 20.8m |
| | CaDA | 18.504 | 1.140% | 3s | 30.072 | 2.044% | 14s |
| | CaDA-EAS | 18.378 | 0.464% | 9.3m | 29.846 | 1.278% | 1.23h |
| | PoMtVRS(CaDA) | 18.478 | 0.995% | 4s | 29.981 | 1.729% | 20s |
| | PoMtVRS(CaDA)-EAS | 18.358 | **0.352%** | 19.1m | 15.768 | **0.926%** | 1.40h |
| | PoMtVRS(CaDA)-EAS-LoRA | 18.386 | **0.502%** | 5.3m | 29.753 | **0.960%** | 57.0m |
| OVRPB | HGS-PyVRP | 6.898 | * | 10.4m | 10.335 | * | 20.8m |
| | OR-Tools | 6.928 | 0.412% | 10.4m | 10.577 | 2.315% | 20.8m |
| | CaDA | 7.047 | 2.129% | 3s | 10.761 | 4.083% | 13s |
| | CaDA-EAS | 6.956 | 0.824% | 8.5m | 10.634 | 2.859% | 1.10h |
| | PoMtVRS(CaDA) | 7.021 | 1.755% | 4s | 10.657 | 3.093% | 17s |
| | PoMtVRS(CaDA)-EAS | 6.939 | **0.581%** | 17.4m | 10.508 | **1.655%** | 1.33h |
| | PoMtVRS(CaDA)-EAS-LoRA | 6.952 | **0.763%** | 8.0m | 10.542 | **1.977%** | 57.2m |
| OVRPBLTW | HGS-PyVRP | 11.668 | * | 10.4m | 19.156 | * | 20.8m |
| | OR-Tools | 11.681 | 0.106% | 10.4m | 19.305 | 0.767% | 20.8m |
| | CaDA | 11.761 | 0.778% | 3s | 19.441 | 1.468% | 14s |
| | CaDA-EAS | 11.711 | 0.362% | 10.0m | 19.328 | 0.639% | 1.20h |
| | PoMtVRS(CaDA) | 11.744 | 0.637% | 5s | 19.395 | 1.224% | 20s |
| | PoMtVRS(CaDA)-EAS | 11.694 | **0.213%** | 20.2m | 19.238 | **0.418%** | 1.46h |
| | PoMtVRS(CaDA)-EAS-LoRA | 11.703 | **0.292%** | 5.1m | 19.251 | **0.487%** | 59.3m |
| OVRPL | HGS-PyVRP | 6.507 | * | 10.4m | 9.724 | * | 20.8m |
| | OR-Tools | 6.552 | 0.668% | 10.4m | 10.001 | 2.791% | 20.8m |
| | CaDA | 6.669 | 2.451% | 3s | 10.127 | 4.100% | 13s |
| | CaDA-EAS | 6.583 | 1.145% | 8.5m | 10.018 | 2.984% | 1.17h |
| | PoMtVRS(CaDA) | 6.639 | 2.004% | 4s | 10.033 | 3.146% | 17s |
| | PoMtVRS(CaDA)-EAS | 6.552 | **0.679%** | 17.2m | 9.915 | **1.929%** | 1.33h |
| | PoMtVRS(CaDA)-EAS-LoRA | 6.566 | **0.889%** | 8.4m | 9.938 | **2.169%** | 55.1m |

| Task | Solver | N=50 Obj. | Gap | Time | N=100 Obj. | Gap | Time |
|---|---|---|---|---|---|---|---|
| VRPTW | HGS-PyVRP | 16.031 | * | 10.4m | 25.423 | * | 20.8m |
| | OR-Tools | 16.089 | 0.347% | 10.4m | 25.814 | 1.506% | 20.8m |
| | CaDA | 16.276 | 1.522% | 3s | 26.083 | 2.579% | 13s |
| | CaDA-EAS | 16.163 | 0.815% | 9.2m | 25.888 | 1.818% | 1.16h |
| | PoMtVRS(CaDA) | 16.236 | 1.272% | 4s | 25.976 | 2.151% | 20s |
| | PoMtVRS(CaDA)-EAS | 16.117 | **0.532%** | 18.4m | 25.763 | **1.317%** | 1.38h |
| | PoMtVRS(CaDA)-EAS-LoRA | 16.140 | **0.674%** | 8.0m | 25.838 | **1.609%** | 39.2m |
| VRPL | HGS-PyVRP | 10.587 | * | 10.4m | 15.766 | * | 20.8m |
| | OR-Tools | 10.570 | 2.343% | 10.4m | 16.466 | 5.302% | 20.8m |
| | CaDA | 10.730 | 1.330% | 3s | 16.061 | 1.875% | 13s |
| | CaDA-EAS | 10.647 | 0.545% | 8.3m | 15.971 | 1.302% | 1.13h |
| | PoMtVRS(CaDA) | 10.698 | 1.028% | 4s | 15.951 | 1.188% | 17s |
| | PoMtVRS(CaDA)-EAS | 10.625 | **0.339%** | 16.5m | 15.873 | **0.690%** | 1.31h |
| | PoMtVRS(CaDA)-EAS-LoRA | 10.637 | **0.456%** | 8.4m | 15.896 | **0.837%** | 51.1m |
| OVRPTW | HGS-PyVRP | 10.510 | * | 10.4m | 16.926 | * | 20.8m |
| | OR-Tools | 10.519 | 0.078% | 10.4m | 17.027 | 0.583% | 20.8m |
| | CaDA | 10.619 | 1.013% | 3s | 17.234 | 1.793% | 14s |
| | CaDA-EAS | 10.561 | 0.472% | 9.4m | 17.131 | 1.189% | 1.24h |
| | PoMtVRS(CaDA) | 10.593 | 0.775% | 4s | 17.175 | 1.443% | 20s |
| | PoMtVRS(CaDA)-EAS | 10.541 | **0.288%** | 19.4m | 17.059 | **0.768%** | 1.43h |
| | PoMtVRS(CaDA)-EAS-LoRA | 10.551 | **0.375%** | 6.1m | 17.086 | **0.924%** | 51.3m |
| VRPBLTW | HGS-PyVRP | 18.361 | * | 10.4m | 29.026 | * | 20.8m |
| | OR-Tools | 18.422 | 0.332% | 10.4m | 29.830 | 2.770% | 20.8m |
| | CaDA | 18.852 | 1.402% | 3s | 30.534 | 2.407% | 14s |
| | CaDA-EAS | 18.714 | 0.663% | 9.3m | 30.295 | 1.610% | 1.24h |
| | PoMtVRS(CaDA) | 18.814 | 1.194% | 4s | 30.429 | 2.048% | 20s |
| | PoMtVRS(CaDA)-EAS | 18.671 | **0.436%** | 19.2m | 30.088 | **0.919%** | 1.41h |
| | PoMtVRS(CaDA)-EAS-LoRA | 18.701 | **0.595%** | 6.4m | 30.145 | **1.107%** | 59.2m |
| VRPLTW | HGS-PyVRP | 16.356 | * | 10.4m | 25.757 | * | 20.8m |
| | OR-Tools | 16.441 | 0.499% | 10.4m | 26.259 | 1.899% | 20.8m |
| | CaDA | 16.670 | 1.890% | 3s | 26.534 | 2.977% | 14s |
| | CaDA-EAS | 16.517 | 0.971% | 9.2m | 26.295 | 2.060% | 1.21h |
| | PoMtVRS(CaDA) | 16.634 | 1.664% | 4s | 26.433 | 2.581% | 20s |
| | PoMtVRS(CaDA)-EAS | 16.461 | **0.630%** | 18.5m | 26.141 | **1.462%** | 1.39h |
| | PoMtVRS(CaDA)-EAS-LoRA | 16.491 | **0.811%** | 8.4m | 26.225 | **1.783%** | 47.1m |
| OVRPBL | HGS-PyVRP | 6.899 | * | 10.4m | 10.335 | * | 20.8m |
| | OR-Tools | 6.927 | 0.386% | 10.4m | 10.582 | 2.363% | 20.8m |
| | CaDA | 7.049 | 2.146% | 3s | 10.762 | 4.093% | 13s |
| | CaDA-EAS | 6.982 | 1.182% | 8.5m | 10.593 | 2.467% | 1.10h |
| | PoMtVRS(CaDA) | 7.022 | 1.755% | 4s | 10.659 | 3.110% | 18s |
| | PoMtVRS(CaDA)-EAS | 6.941 | **0.591%** | 17.5m | 10.480 | **1.381%** | 1.33h |
| | PoMtVRS(CaDA)-EAS-LoRA | 6.954 | **0.772%** | 8.1m | 10.496 | **1.535%** | 1.00h |
| OVRPBTW | HGS-PyVRP | 11.669 | * | 10.4m | 19.156 | * | 20.8m |
| | OR-Tools | 11.682 | 0.109% | 10.4m | 19.303 | 0.757% | 20.8m |
| | CaDA | 11.759 | 0.757% | 3s | 19.441 | 1.472% | 14s |
| | CaDA-EAS | 11.709 | 0.337% | 10.0m | 19.320 | 0.842% | 1.19h |
| | PoMtVRS(CaDA) | 11.744 | 0.634% | 5s | 19.394 | 1.222% | 20s |
| | PoMtVRS(CaDA)-EAS | 11.695 | **0.219%** | 20.2m | 19.247 | **0.465%** | 1.46h |
| | PoMtVRS(CaDA)-EAS-LoRA | 11.703 | **0.285%** | 6.2m | 19.268 | **0.574%** | 59.5m |
| OVRPLTW | HGS-PyVRP | 10.510 | * | 10.4m | 16.926 | * | 20.8m |
| | OR-Tools | 10.497 | 0.114% | 10.4m | 17.023 | 0.728% | 20.8m |
| | CaDA | 10.620 | 1.021% | 3s | 17.234 | 1.793% | 14s |
| | CaDA-EAS | 10.563 | 0.491% | 9.4m | 17.128 | 1.173% | 1.25h |
| | PoMtVRS(CaDA) | 10.593 | 0.776% | 4s | 17.174 | 1.440% | 20s |
| | PoMtVRS(CaDA)-EAS | 10.541 | **0.288%** | 19.3m | 17.047 | **0.700%** | 1.42h |
| | PoMtVRS(CaDA)-EAS-LoRA | 10.550 | **0.364%** | 6.2m | 17.081 | **0.897%** | 46.3m |

