# OpenReview forum: "PoMtVRS: Preference-Optimized Multi-Task Vehicle Routing Solver with Preference Gating"
_ICML.cc/2026/Conference — ICML 2026 regular_

### Official Review · Reviewer_WuhV · 2026-02-20

**Soundness:** 3
**Presentation:** 3
**Significance:** 3
**Originality:** 3
**Overall Recommendation:** 5
**Confidence:** 4

**Summary:**

This paper studies preference-based multi-task learning for neural VRP solvers and proposes PoMtVRS, featuring a Preference-Gated Block (PGB) and Preference Optimization (PO). PGB injects state information into decoding, while PO updates the model using within-instance solution rankings rather than baseline-based rewards, aiming to reduce reward-scale and noise mismatch across variants. Experiments support the effectiveness of the approach.

**Compliance With Llm Reviewing Policy:**

Affirmed.

**Final Justification:**

I find the paper to be sound and reasonably original, with practical significance for multi-task VRP learning. The proposed method is well motivated, and the empirical results are strong. The rebuttal addressed my concerns. My final evaluation is more positive than before.

**Key Questions For Authors:**

1. Why does head-wise gating improve performance, and what is the conceptual connection between PGB and “preference”?
2. Is there a reason that the ablation version w.o. Map (remove the mapping in PGB while keeping PO) is infeasible? If no, how does it perform?
3. How does the reward difference between multiple solutions of the same instance vary across tasks/variants under standard baselines?

**Limitations:**

It would be helpful to discuss limitations of PO. While PO reduces noise by learning from rankings instead of raw rewards, it also discards the magnitude of quality differences between solutions on the same instance. Does this imply that when reward scales are already comparable (e.g., fixed node size and mild constraint-induced solution performance shifts), PO may learn less efficiently than standard RL methods that exploit reward magnitudes?

**Strengths And Weaknesses:**

Strengths:
- The paper is well structured and easy to follow.
- The experimental results provide solid support for the main claims.
- The proposed components appear easy to integrate into existing solvers.
- Involving PO is an interesting idea that can reduce training noise caused by inconsistent reward/objective scales across variants.

Weaknesses:
- The rationale for PGB, especially head-wise gating, is insufficient: why does head-wise gating improve performance, and what is the conceptual connection between PGB and “preference”?
- The method description lacks clarity in several places:
  - $h_o$ and $h_c$ in Fig. 2 do not seem to be defined.
  - The exact form of the reshaping function $\mathcal{F}$ is not explained.
  - The state $s_t$ is not clearly defined (how it is constructed from $\mathcal{G}$ and the partial solution).
- PO is motivated by the claim that variant-dependent reward scales in multi-task learning lead to high-variance gradients and inefficient exploration, but direct empirical evidence that this issue actually occurs is limited. It would strengthen the motivation to explicitly visualize/quantify, during training, how the reward difference between multiple solutions of the same instance varies across tasks/variants under standard baselines.
- A key ablation appears missing: w.o. Map (remove the mapping in PGB while keeping PO) to isolate the contribution of the mapping module. Is there a reason this ablation is infeasible?
- Minor notation issue: in Sec. 4.3, a policy maps states to actions, while $p_\theta(\tau|\mathcal{G})$ reads more like a probability than a policy.

---

> ### Author Rebuttal · Authors · 2026-03-31
>
> > **Q1, W1: Head-wise gating**
>
> A1: Thank you for the question.
> (1)Head-wise gating improves performance by enabling head-specific, state-conditional reweighting of decoder context, which is crucial in multi-task VRPs where changing constraints alter the effective decision rule across tasks and decoding steps. PGB combines head-wise gating, state mapping, and nonlinear residual refinement to produce a preference-enhanced decoding representation, rather than relying only on fixed shared context embeddings. This is supported by ablations: removing only gating (w.o.Gate) degrades performance from 2.205% to 2.241% on N100, while still outperforming removing PGB (w.o.block), showing that gating provides a clear gain beyond the other PGB components.
>
> (2)"Preference" is twofold. Optimization-wise, PO converts within-instance route comparisons into preference supervision based on relative, not absolute rewards. Architecturally, PGB makes the decoder representation preference-aware by learning which head-wise contextual pathways should be emphasized under the current state. Thus, PGB provides the representation mechanism, while PO provides the matched supervision signal.
> > **Q2, W4: Ablation**
>
> A2: The w.o. Map variant is fully feasible. We run this ablation, and the results show that w.o. Map is better than CaDA backbone but worse than PoMtVRS, indicating that state attribute mapping is not redundant. It provides an additional gain by explicitly injecting state/constraint information into decoder representations.
> Avg.Gap(%)|CaDA|w.o.block|w.o.Gate|w.o.Map|PoMtVRS
> -|-|-|-|-|-
> N50|1.721|1.579|1.427|**1.423**|1.405
> N100|2.844|2.529|2.241|**2.323**|2.205
>
> We did not report w.o.Map because we previously grouped mapping and nonlinear refinement together as the non-gating part and chose only some representative ablations. We agree that this variant is important for completeness and will include it explicitly in the revision.
> > **Q3, W3, Limitations: Reward differences**
>
> A3: Thank you for helpful suggestions. We fully agree it. (1)For PoMtVRS, the issue is not only different absolute reward scales, but that the distribution of reward gaps among multiple solutions of the same instance is heterogeneous across variants and progressively contracts during training. Since REINFORCE relies directly on scalar reward/advantage signals, this yields uneven gradient scales across variants and weaker optimization signals later in training. In contrast, PO maps reward differences into pairwise preference distributions and depends mainly on relative ordering rather than raw reward magnitude, which makes it more stable.
> We additionally sampled multiple solutions per instance under a standard RL baseline and collected the corresponding advantage scales. The statistics show that the RL advantage distribution is concentrated near 0-1. By contrast, PO distributions are much broader, implying stronger discrimination between better and worse candidates. We agree that the evidence is still indirect, and in the revision we will further add explicit diagnostics under standard baselines, such as the mean pairwise reward gap and max–min reward spread across variants and training stages, to directly show both the cross-variant heterogeneity of reward differences and their progressive contraction.
>
> (2) Regarding limitations, we agree that PO may discard some useful fine-grained magnitude information. In principle, if reward scales are already well calibrated and highly comparable, standard RL may exploit them more directly than PO. However, the results suggest that it is not the dominant regime here. On N=50, reward normalization improves RL from 1.621% to 1.569%, confirming that reward scale matters. But w. PO-Norm still reaches 1.418%, and full PoMtVRS further improves to 1.405%. This indicates that PO’s gain is not only from mitigating reward-scale mismatch, but also from providing more stable within-instance preference supervision than magnitude-based RL. Thus, while this limitation exists in principle, in the multi-task VRP setting studied here the stability benefit of PO outweighs the loss of raw magnitude information. We will add this discussion explicitly in the revision.
> N50|w.RL-Norm|w.PO-Norm|w.RL|PoMtVRS
> -|-|-|-|-
> Avg.Obj.|11.499|11.483|11.503|11.482
> Avg.Gap(%)|1.569|1.418|1.621|1.405
>
> > **W2, W5: Notation**
>
> A4: Thanks for pointing these out. We will revise the notation in the final version.
>
> (1)$h_o$ denotes the MHA output in Sec.3.2, i.e., $M_t^{H}=\mathrm{Concat}(z_t^{(1)},\ldots,z_t^{(H)})$, while $h_c$ denotes context embeddings used to condition PGB.
>
> (2)$\mathcal{F}$ is an inverse-concatenation operator that splits $M_t^H\in\mathbb{R}^{d}$ into head-wise blocks in $\mathbb{R}^{H\times d_h}$.
>
> (3)$s_t$ denotes the dynamic state determined by $\mathcal{G}$ and partial solution $\tau_{<t}$. For CaDA, $s_t=[c_t^{LH},c_t^{BH},t_t,\ell_t,o_t]$.
>
> (4)We will revise the notation to a distinct symbol, e.g.,$\pi_\theta(\tau|\mathcal{G})$.

---

> > ### Author Rebuttal · Reviewer_WuhV · 2026-04-03
> >
> > Thank you for the thorough rebuttal and the additional experimental results. The rebuttal has addressed my concerns. I will increase my score.

---

> > > ### Author Response · Authors · 2026-04-03
> > >
> > > We are pleased that our rebuttal has fully addressed all of your concerns, and we sincerely appreciate your decision to increase your score. Thank you for your thorough and constructive feedback throughout the review process, which has been instrumental in refining the quality of our manuscript.

---

### Official Review · Reviewer_Xew2 · 2026-02-25

**Soundness:** 3
**Presentation:** 3
**Significance:** 3
**Originality:** 2
**Overall Recommendation:** 4
**Confidence:** 3

**Summary:**

This paper proposes PoMtVRS, a plug-and-play framework for multi-task vehicle routing problems (VRPs) that combines architectural augmentation with preference-driven optimization. The authors address two key challenges in existing neural VRP solvers: (1) insufficient representation and unstable training under constraint switching, and (2) inefficient exploration in large combinatorial action spaces. The solution consists of two main components: a preference-gated block (PGB) that adaptively modulates decoder representations through head-wise gating and nonlinear refinement, and a preference optimization (PO) objective that learns relative comparisons among candidate solutions rather than relying on absolute rewards. The framework is evaluated on 16 VRP variants and demonstrates consistent improvements over state-of-the-art baselines when integrated with existing backbones (MVMoE and CaDA).

**Compliance With Llm Reviewing Policy:**

Affirmed.

**Key Questions For Authors:**

1. What is the computational overhead of the preference-gated block during inference, and how does it scale with problem size? Could this become a bottleneck for very large-scale VRPs?
2. The authors claim to outline a central aspect of multi-task VRP solving through preference optimization, but how sensitive is the method to the choice of hyperparameters, particularly the temperature coefficient α? Is there a principled way to set this parameter?
3. Could the preference-gated block architecture be applied to other combinatorial optimization problems beyond VRPs? What aspects of the design are specific to routing problems?
4. How does PoMtVRS compare to specialized single-task solvers when evaluated on individual VRP variants? Is there a performance trade-off for the multi-task capability?

**Limitations:**

Yes

**Strengths And Weaknesses:**

Strengths:
1. The paper includes extensive experiments across 16 VRP variants, real-world CVRPLib instances, and zero-shot generalization to unseen constraints.
2. PoMtVRS can be seamlessly integrated with existing neural solvers, demonstrating practical utility and versatility.
3. Consistent improvements across all evaluated variants and scales, with particularly notable gains on larger problem instances.
4. The method shows robust performance on out-of-distribution instances and unseen constraint combinations.

Weaknesses:
1. While the empirical results are strong, the paper lacks deeper theoretical justification for why the preference-gated block and preference optimization work well together. The authors address the concept of preference optimization but provide minimal theoretical grounding for their architectural choices.
2. The additional complexity introduced by the preference-gated block is not thoroughly analyzed in terms of inference-time overhead and scalability to much larger problem instances.
3. Limited discussion of how sensitive the method is to key hyperparameters, particularly the temperature coefficient α in the preference optimization objective.

---

> ### Author Rebuttal · Authors · 2026-03-31
>
> > **Q1, W2: PGB overhead**
>
> A1: PGB is a lightweight decoder-side module. It adds neither extra cross-node attention nor extra search iterations, so it only increases modest inference overhead without changing the dominant scaling of autoregressive decoding. Under the standard protocol(1K instances per task), inference remains in the seconds regime. At N=100 w.o. block vs. PoMtVRS is about 14\~17s vs. 17\~20s, corresponding to only a few extra milliseconds per instance.
> Gap|w.o.block|PoMtVRS
> -|-|-
> N50|1.579%(3~4s)|1.405%(4~5s)
> N100|2.529%(14~17s)|2.205%(17~20s)
>
> Although some overhead is unavoidable as size grows, the results suggest PGB is unlikely to be a major bottleneck. Appendix C.4 shows that on CVRPLIB 101–1001 nodes, PoMtVRS achieves better solution quality with practical runtime relative to CaDA. Moderate inference overhead is often acceptable in VRP when it yields better solutions. Compared with refinement-based methods (EAS), whose runtime may rise from seconds to minutes or hours, PGB adds much smaller overhead. Overall, we believe the added cost of PGB is modest and acceptable and unlikely to be a bottleneck for large-scale VRPs.
> > **Q2, W3: Hyperparameter sensitivity**
>
> A2: We agree it. In PO, α comes from the entropy-regularized objective and controls the exploration–exploitation trade-off. Prior work[5] provides a principled empirical guideline. α should be chosen according to the solver’s inherent exploration capacity and the problem. For VRP, effective values are typically in the $10^{-2}\-10^{-3}$, and [5] identifies α=0.03 as a better choice for CVRP. Following that, we evaluate {0.001,...,0.1,1}. The best result is α=0.03. Once the backbone and problem class are fixed, a small local search suffices. We will add this study in the revision.
> α|0.001|0.003|0.005|0.01|0.03|0.05|0.08|0.1|1
> -|-|-|-|-|-|-|-|-|-
> Avg.Obj|11.489|11.495|11.493|11.491|11.482|11.483|39.18|39.18|20.78
> Avg.Gap(%)|1.450|1.527|1.513|1.496|**1.405**|1.413|258.95|258.95|78.01
>
> [5]Preference Optimization for Combinatorial Optimization Problems, ICML 2025.
> > **Q3: Transferability of PGB**
>
> A3: Multi-task VRP is itself a meaningful and still under-explored setting. Following prior works, we focus on it and achieve strong performance across all 16 VRPs, surpassing existing SOTA multi-task baselines. We additionally applied PGB and PoMtVRS to Flexible Flow Shop Scheduling Problem (FFSP). The results show consistent improvements on both FFSP20/50, indicating that they are not limited to VRPs. The main routing-specific part is state attribute mapping, which currently uses routing-related constraints/states. For other COPs, it can be naturally replaced by task-specific feasibility or state descriptors, while keeping the architecture unchanged.
> ||FFSP20|FFSP50
> -|-|-
> MatNet|25.388(1.003%)|49.642(0.637%)
> PGB(MatNet)|25.277(0.561%)|49.540(0.430%)
> PoMtVRS(MatNet)|25.136(0.00%)|49.328(0.00%)
> > **Q4: Single-task solvers**
>
> A4: Prior multi-task works(our baselines) usually compare only with other multi-task solvers(MS), since comparisons with separate single-task solvers are not fully like-for-like and multi-tasks are generally disadvantaged. We additionally compared PoMtVRS with specialized single-task solvers(SS). As expected, like other multi-task solvers, PoMtVRS below single-task solvers in many cases, but it can surpass them in some cases while consistently outperforming current SOTA multi-task solvers.
> ||CVRP50||VRPB50
> -|-|-|-
> POMO(SS)|10.463(0.875%)|POMO|14.162(35.13%)
> LEHD RRC50(SS)|10.455(0.804%)|DRL-VRPB(SS)[6]|13.056(24.58%)
> CADA(MS)|10.495(1.186%)|CaDA|9.960(2.798%)
> PoMtVRS|10.459(0.843%)|PoMtVRS|9.899(2.171%)
>
> The trade-off is inherent to multi-task solvers in general. The more variants a unified solver covers, the harder it is to match specialized per-variant models. Still, PoMtVRS narrows this gap substantially. It improves its backbone on all 16 VRPs, and in some cases even exceeds specialized single-task models. The trade-off is between unified modeling and fully specialized modeling, and PoMtVRS significantly reduces that gap while enhancing multi-task capability.
>
> [6]Deep Reinforcement Learning for Solving Vehicle Routing Problems With Backhauls, TNNLS 2025.
> > **W1: Architectural choices**
>
> A5: PGB improves decoder expressiveness via state-conditioned modulation, while PO stabilizes optimization by replacing raw scalar rewards with within-instance preference pairs. Thus, PGB provides the representation capacity to express state-dependent priorities, and PO provides the matched supervision signal to rank better candidate routes higher. Although we do not formalize this as a theorem, the evidence supports it: with CaDA at N=100, replacing RL with PO improves performance from 2.844% to 2.529%, and adding PGB further improves it to 2.205%. Removing either PO or PGB degrades performance. This suggests that the gain comes from their deliberate representation–optimization alignment, which we will clarify in the revision.

---

### Official Review · Reviewer_He7o · 2026-03-12

**Soundness:** 3
**Presentation:** 2
**Significance:** 2
**Originality:** 2
**Overall Recommendation:** 4
**Confidence:** 2

**Summary:**

This paper proposes PoMtVRS, which introduces a preference vector to represent the combination of constraints for the current task, thereby enabling a unified framework to solve multiple VRP variants.

**Compliance With Llm Reviewing Policy:**

Affirmed.

**Final Justification:**

I have no further questions and I will maintain my original score.

**Key Questions For Authors:**

1. See ``W1-W2``

**Limitations:**

yes

**Strengths And Weaknesses:**

## Strength
1. The paper considers 16 variants of the VRP, conducts extensive experiments, and achieves promising performance. The manuscript is concise and clearly written.
2. To address the issues of inconsistent reward scales across multiple tasks and low exploration efficiency, the paper adopts a BT-based preference optimization strategy, which effectively alleviates the gradient interference caused by large scale differences among different tasks.

## Weakness

1. The experiments in this paper focus on problem sizes of 50 and 100, while larger-scale instances, such as 200 and 500, are not further investigated.
2. As shown in Table 4, the proposed method shows some improvement under unseen constraints; however, the overall performance remains relatively limited.

---

> ### Author Rebuttal · Authors · 2026-03-31
>
> Thank you for your constructive and valuable comments and suggestions. We address your concerns point-by-point as follows:
>
> > **Q1: Larger-scale evaluation.**
>
> **A1:** Thank you for raising this point. We agree that evaluating on larger-scale instances (e.g., 200, 500, or beyond) is important. Our main experiments focus on N=50,100 because this follows the standard protocol used in recent unified multi-task neural VRP solvers[1,2,3,4] and provides a fair common setting for all neural baselines considered in our paper. We would also like to emphasize that multi-task VRP is already a highly challenging setting. Unlike the setting of a single VRP, in our multi-task setting a single model must simultaneously handle 16 compositional VRP variants with substantially different feasibility structures, state attributes, and routing behaviors. Improving performance on problem scales like N=50,100 usually becomes an important basis for developing more advanced methods for dealing with large-scale instances.
>
> That said, PoMtVRS is not limited to the small-scale regime. In fact, the paper already includes additional evaluation on larger real-world-style instances from CVRPLIB datasets, which contains 100 instances ranging from 101 to 1001 nodes. The main text (Table 3) reports size-binned average gaps, while Appendix C.4 provides instance-wise comparisons. Importantly, PoMtVRS(CaDA) consistently outperforms the baselines across all three size ranges. In particular, for 251≤N<501, PoMtVRS achieves 7.865%, improving over 8.889% from CaDA. For the even larger regime 501<N≤1001, our method reaches 12.651%, outperforming 14.199% from CaDA and 12.814% from MoSES(CaDA). These results indicate that, although training is conducted under the N=100 setting, PoMtVRS still generalizes well to substantially larger instances, including around 500 nodes and even up to 1000 nodes.
> CVRPLIB(Avg.Gap)|MTPOMO|MVMoE|RF-TE|CaDA|MoSES(CaDA)|PoMtVRS(CaDA)
> -|-|-|-|-|-|-
> N<251|6.566%|5.829%|5.061%|4.772%|4.724%|4.561%
> 251≤N<501|11.529%|10.616%|8.107%|8.889%|7.948%|7.865%
> 501<N≤1001|30.190%%|18.918%|12.253%|14.199%|12.814%|12.651%
>
> We appreciate the suggestion and will make this point clearer in the final version. We also agree that unified multi-task VRP at larger scales is itself an important future direction, and it is one of the key directions we plan to pursue next.
>
> > **Q2: Unseen constraints performance.**
>
> **A2:** Thank you for this helpful comment. We agree that the overall performance under unseen constraints in Table 4 still leaves room for improvement. We would like to clarify that this is a particularly challenging setting. The model is trained only on the 16 compositional variants derived from the base constraints, but at testing it is required to generalize zero-shot to new constraint mechanisms and their compositions, including MD, MB, and MDMB, covering 32 unseen VRP variants in total. We also explicitly point this out as an important remaining challenge in the main text.
>
> We would also like to emphasize that most neural VRP methods struggle to zero-shot generalization with unseen constraints. Instead, they are typically trained and evaluated only on a single VRP variant. Only a few recently proposed unified multi-task solvers[1,2,3,4], such as our baselines MTPOMO, MVMoE, RouteFinder, and CaDA, are suitable for meaningful comparison in this context. Notably, even these strong multi-task baselines exhibit relatively large absolute gaps under unseen constraints, which reflects the intrinsic difficulty of the problem rather than a limitation specific to our method.
>
> Despite this difficulty, our PoMtVRS achieves the best results across all 32 unseen constraint types and both scales (VRP50/100). In particular, under the most challenging MDMB setting, our method reduces the gap by 10.7% on VRP50. It also consistently outperforms all compared methods. These results suggest that, although the absolute performance under unseen constraints is still limited, our method substantially improves generalization in this highly challenging unified multi-task zero-shot setting and establishes the strongest performance among the compared baselines.
>
> We appreciate the reviewer for highlighting this issue. We agree that improving generalization of multi-task solvers to unseen constraints and their compositions remains an important open problem, and this is fully aligned with the limitations and future directions discussed in our paper.
>
> [1] MVMoE: Multi-Task Vehicle Routing Solver with Mixture-of-Experts, ICML 2024.
>
> [2] ROUTEFINDER: Towards Foundation Models for Vehicle Routing Problems, TMLR 2025.
>
> [3] CaDA: Cross-Problem Routing Solver with Constraint-Aware Dual-Attention, ICML 2025.
>
> [4] Multi-Task Vehicle Routing Solver via Mixture of Specialized Experts under State-Decomposable MDP, NeurIPS 2025.

---

> > ### Author Rebuttal · Reviewer_He7o · 2026-04-03
> >
> > I have no further questions and I will maintain my original score.

---

> > > ### Author Response · Authors · 2026-04-04
> > >
> > > We are delighted that our response has addressed your concerns, and we greatly thank you for your time and effort in evaluating the work.

---

### Decision · Program_Chairs · 2026-04-30

**Decision:**

Accept (regular)

**Comment:**

PoMtVRS proposes a plug-and-play multi-task VRP framework that addresses insufficient decoder representation and cross-variant reward scale inconsistency via a Preference-Gated Block (PGB) and Preference Optimization (PO), achieving consistent improvements across 16 VRP variants and real-world CVRPLIB instances. All three reviewers were broadly positive (4/4/5), and the rebuttal provided useful additional experiments including PGB inference overhead, hyperparameter sensitivity analysis, and the missing w.o.Map ablation; Reviewer WuhV raised their score to 5 post-rebuttal while He7o maintained 4 and Xew2 did not update. The main limitations are relatively weak theoretical justification, limited originality (both PGB and PO adapt existing ideas), and insufficient evaluation on very large instances (500+ nodes). Nevertheless, as a practically useful framework that integrates seamlessly with existing solvers and demonstrates solid empirical coverage, the paper meets the bar for acceptance.